# FcRn-silencing of IL-12Fc prevents toxicity of local IL-12 therapy and prolongs survival in experimental glioblastoma

Michal Beffinger [1,2,13], Linda Schellhammer [1,13], Betül Taskoparan [1,13], Sereina Deplazes [1,2], Ulisse Salazar [1], Nazanin Tatari [3], Frauke Seehusen[4], Leopold von Balthazar[5], Carl Philipp Zinner [1,6], Sabine Spath [2], Tala Shekarian[3], Marie-Françoise Ritz [3], Marta McDaid[3], Pascal Egloff[7], Iwan Zimmermann [7,12], Hideho Okada [8,9], E. Sally Ward[10], Jack Rohrer[5], Markus A. Seeger [7], Thorsten Buch [1], Gregor Hutter [3,11] & Johannes vom Berg [1,2]✉

Glioblastoma remains a challenging indication for immunotherapy: the blood-brain barrier hampers accessibility for systemic treatments and the immuno-suppressive microenvironment impedes immune attack. Intratumoral therapy with the proinflammatory cytokine interleukin-12 (IL-12) can revert immuno-suppression but leakage into the circulation causes treatment-limiting toxicity. Here we engineer an IL-12Fc fusion cytokine with reduced binding to the neonatal Fc receptor FcRn. FcRn-silenced IL-12Fc avoids FcRn-mediated brain export, thus exhibits prolonged brain retention and reduced blood levels, which prevents toxicity. In murine glioblastoma, FcRn-silenced IL-12Fc induces more durable responses with negligible systemic cytokine exposure and boosts the efficacy of radio- and chemotherapy. It triggers anti-tumor responses independently of peripheral T cell influx or lymphopenia and leads to inflammatory polarization of the tumor microenvironment in patient-derived glioblastoma explants. FcRn-silencing of IL-12Fc may unlock the full potential of IL-12 for brain cancer therapy and could be further applied to containing the activity of other therapeutics targeting neurological diseases.

Glioblastoma (GBM) is the most common and aggressive tumor of the central nervous system in adults. Despite refinements in surgical resection and chemotherapy, and radiotherapy combination regimens[1], GBM carries a median survival time of less than 2 years after diagnosis[2]. While therapy with immune checkpoint inhibitors (ICIs) has revolutionized the prognosis of patients with other types of solid tumors[3,4], GBM is characterized by low immunogenicity, severe immunosuppression, and marked immune evasion, rendering it

[1]Institute of Laboratory Animal Science, University of Zurich, Schlieren, Switzerland. [2]InCephalo AG, Allschwil, Switzerland. [3]Brain Tumor Immunotherapy and Biology Lab, Department of Biomedicine, University Hospital and University of Basel, Basel, Switzerland. [4]Laboratory for Animal Model Pathology (LAMP), Institute of Veterinary Pathology, University of Zurich, Zurich, Switzerland. [5]Institute of Chemistry and Biotechnology, Zurich University of Applied Sciences, Waedenswil, Switzerland. [6]Institute of Medical Genetics and Pathology, University Hospital Basel, Basel, Switzerland. [7]Institute of Medical Microbiology, University of Zurich, Zurich, Switzerland. [8]Department of Neurological Surgery, University of California, San Francisco, CA, USA. [9]Parker Institute for Cancer Immunotherapy, San Francisco, CA, USA. [10]Cancer Sciences Unit, Centre for Cancer Immunology, University of Southampton, Southampton, UK. [11]Department of Neurosurgery, University Hospital of Basel, Basel, Switzerland. [12]Present address: Linkster Therapeutics AG, Zurich, Switzerland. [13]These authors contributed equally: Michal Beffinger, Linda Schellhammer, Betül Taskoparan. ✉e-mail: johannes.vomberg@uzh.ch

intrinsically resistant to immunotherapy[5]. The blood-brain barrier (BBB) poses additional challenges by limiting access of potentially therapeutic monoclonal antibodies (mAbs): reaching sufficiently high local concentrations at systemically tolerable doses is challenging. For example, in the CHECKMATE-143 trial, the combination of anti-CTLA-4 and anti-PD-1 mAbs induced treatment-limiting side effects in up to one-third of patients, but showed promise of disease stabilization[6]. More recently, direct, perioperative intracerebral administration of CTLA-4 and PD-1 ICI together with systemic injection of PD-1 ICI improved tolerability and showed an encouraging increase in overall survival[7], demonstrating the potential of local intralesional administration to increase efficacy and tolerability. A major remaining challenge preventing the realization of the potential of immunotherapy in GBM is how to increase the short local residence time of the injected mAbs[8], while preventing systemic toxicity. Alongside, reshaping the immunosuppressive tumor microenvironment (TME) could unlock a tumor-clearing immune response and foster durable responses.

Interleukin 12 (IL-12) is a pro-inflammatory cytokine that stimulates type 1 immunity and is an excellent candidate for inflammatory reprogramming of the TME. IL-12 treatment is effective in many types of experimental tumors, including gliomas[9], but the systemic treatment of patients leads to severe toxicity, mainly provoked by high IFNγ levels in circulation[10,11]. How to reach sufficient intratumoral (i.t.) doses while avoiding systemic side effects has been evaluated in various experimental and clinical settings[12], including GBM: Chiocca et al. used local injection of a ligand-inducible, adenoviral-delivered expression cassette (Ad−RTS−hIL-12) during surgical resection of GBM in patients, but saw immediate leakage of IL-12 into the blood that caused treatment-limiting systemic toxicity[13]. While increased i.t. lymphocytes and IFNγ levels were encouraging signs of bioactivity, the full potential

of local GBM therapy with IL-12 is currently not exploitable due to its poor CNS retention.

In this study, we aimed to overcome both the immunosuppressive TME in GBM and the challenge of brain retention of an immunomodulatory molecule. We engineered a large homodimeric single-chain IL-12Fc fusion protein for local application and identified a dominant role of the neonatal Fc receptor (FcRn) in driving brain-to-blood antibody export[8]. Using murine and human models and primary GBM tumor explants, we reveal the therapeutic potential of FcRn-silenced IL-12Fc to improve survival−both alone and in combination with radio or chemotherapy−with prolonged tissue retention and in the absence of relevant toxicity.

## Results

### Fc-fusion of IL-12 reduces systemic exposure early after CNS administration

A major contributor to tumor retention of therapeutic molecules is their size[14]. Fc-fusion is one way to increase size and can improve physicochemical properties compared to unfused therapeutic candidate molecules[15]. We therefore constructed human IL-12Fc by connecting the IL-12 p35 and p40 subunits with a flexible $(G_4S)_3$ linker, then fusing the p35 subunit to the hinge region of the Fc crystallizable fragment of a human immunoglobulin G4 (Fig. 1A), to generate a homodimer with a predicted molecular mass of 169 kDa (Fig. 1B, C), and low probability of inducing antibody-dependent cell-mediated cytotoxicity or complement-dependent cytotoxicity[16].

To compare the retention of recombinant human IL-12 (rIL-12) versus the IL-12Fc fusion protein, we used a convection-enhanced delivery (CED) protocol adapted to murine brain infusion[17] (Fig. 1D) to administer the cytokines into the parenchyma of hFcRn Tg32 mice.

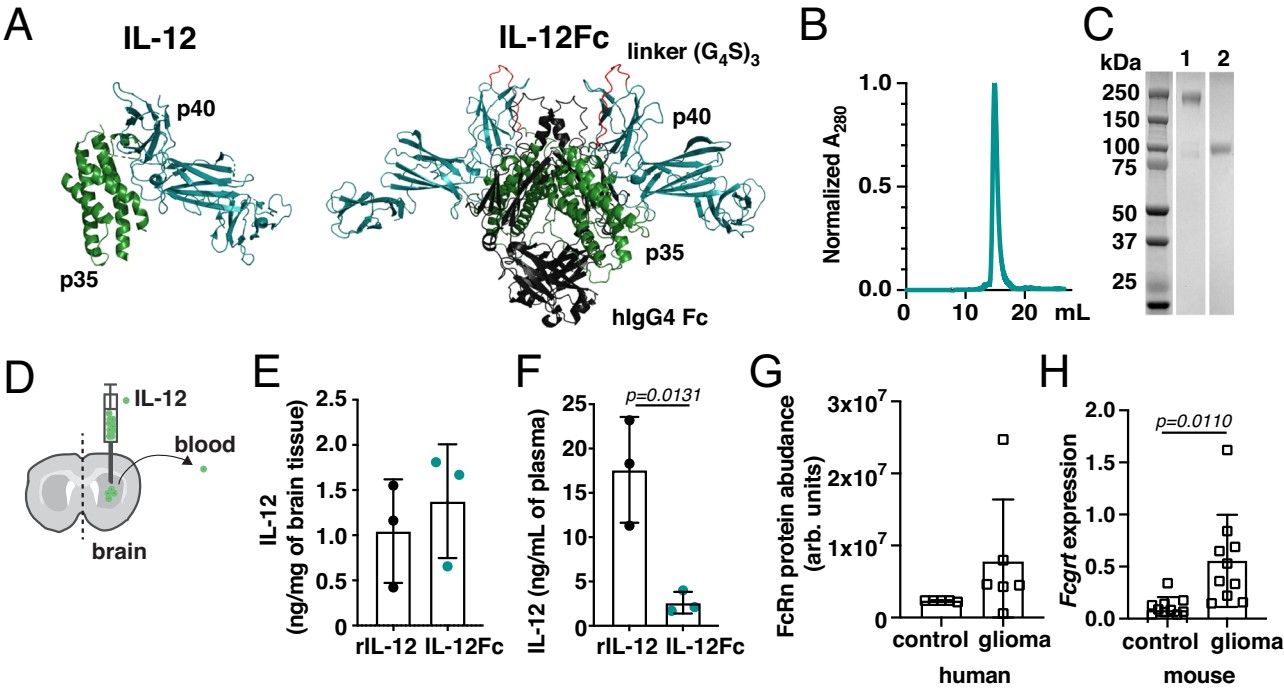

**Fig. 1 | Bivalent single-chain Fc-fusion reduces plasma prevalence of IL-12, but increased FcRn abundance in glioblastoma (GBM) may counteract tissue retention. A** Cartoon of IL-12 (Protein Data Bank entry 1F45)[82] and predicted structure of IL-12Fc[83]. **B** Size exclusion chromatogram (normalized absorbance at 280 nm; $A_{280}$) and **C** SDS-PAGE of IL-12Fc protein in native (lane 1) and reducing conditions (lane 2). Lanes cropped from a single image. Data representative of three independent experiments. **D−F** 1 µg of rIL-12 or IL-12Fc was administered into the right striatum of hFcRn Tg32 mice via convection-enhanced delivery (CED). IL-12 brain levels in the ipsilateral brain hemisphere (**E**) and in plasma (**F**) 6 h after

injection. Unpaired two-tailed $t$-test. Mean ± SD. IL-12Fc WT: $n = 3$ (green), rIL-12: $n = 3$ animals (black). **G** FcRn protein abundance in human brain tissue of non-related disease vs glioma according to the dataset from ref. 20. Unpaired two-tailed $t$-test. Mean ± SD. Control: $n = 5$; glioma: $n = 6$ patients. **H** Relative expression of *Fcgrt* in GL-261 tumor-bearing hemisphere (day 21 post-implantation) and corresponding healthy contralateral hemisphere plotted as fold change of *Fcgrt* to reference gene *Hprt* ($2^{-\Delta Ct}$). $n = 10$ animals/group. Paired two-tailed $t$-test. Mean ± SD. Source data are provided as a Source Data file.

These mice express human FcRn in a pattern typical of wild-type (WT) mice and humans[18] on a murine FcRn-deficient background, and accurately recapitulate the in vivo behavior of antibodies in humans[19]. Compared to rIL-12, IL-12Fc concentration in the brain was slightly higher 6 h after administration (Fig. 1E), while the concentration of IL-12Fc in the blood was significantly (nearly 7-fold) lower at this early timepoint (Fig. 1F). Extending our analysis from 1 μg to a range of doses and timepoints, we found that at a dose of 0.2 μg, rIL-12 levels in plasma were 17-fold lower at 6 h and 7-fold lower at 24 h following treatment with IL-12Fc compared to rIL-12 (Fig. S1A, C); if we administered a 5 μg dose we still observed a significantly lower level of IL-12 in plasma after IL-12Fc administration, albeit only 4-fold reduction in rIL-12 plasma levels at 6 and 24 h (Fig. S1B, D), suggesting an upper dosing threshold for reducing peripheral IL-12 exposure with IL-12Fc. However, despite consistently lower plasma levels, we could not detect any significant increase in brain levels of IL-12Fc (Fig. 1E and S1A-D).

### Brain-expressed neonatal Fc receptor limits local containment of IL-12Fc

Given the lower levels of IL-12Fc than rIL-12 in the plasma, we next asked why the difference in local levels in the brain was not greater. We hypothesized that IL-12Fc could be exported across the BBB through interacting with FcRn[8], though little is known about the expression of FcRn in the context of brain cancer. Therefore, we first analyzed a proteomic screen comparing human glioma versus non-related disease (epilepsy) samples[20], which suggested increased expression of FcRn in glioma (Fig. 1G). Publicly available histology and transcriptome databases showed that FcRn is expressed both within the brain tumor parenchyma and on the vasculature (Fig. S1E–G). In addition, we observed a significant increase in *Fcgrt* expression in the affected compared to the unaffected hemisphere of mice bearing orthotopic GL-261 gliomas (Fig. 1H). Consistent with the evidence of both parenchymal and vascular expression of FcRn, in both human and murine glioma, there was only a weak correlation between the expression of FcRn and the endothelial marker CD31 (Fig. S1H–K).

Increased FcRn expression in brain tumors may facilitate brain-to-blood export of antibodies and therefore could diminish the potential impact of molecular size on improving tissue retention. In the periphery, FcRn-mediated recycling may then cause accumulation of IL-12Fc in the blood over time. To evaluate this possibility, we used osmotic mini-pumps to continuously infuse a murine IL-12Fc (mIL-12Fc) construct[21,22] into the brains of hFcRn Tg32 (in which the hFcRn they express has low affinity for murine Fc[23], and therefore mIL-12Fc) and WT mice bearing GL-261 gliomas, and monitored serum levels of IL-12 over the following 13 days. Increasing levels of IL-12 were detectable in the serum of WT mice from 7 days after pump implantation, whereas in hFcRn Tg32, IL-12 was not detected (Fig. S1L). Together, our findings confirm that the FcRn:IL-12Fc interaction is likely involved in brain-to-blood transcytosis and subsequent blood accumulation of i.t. administered IL-12Fc.

### FcRn-silenced IL-12Fc permits high local dosing with minimal systemic leakage

Considering the increased abundance of FcRn in glioma, its role in IL-12Fc accumulation in blood, and the lower-than-expected difference in local retention of IL-12Fc compared to rIL-12 (Fig. 1E), we next generated variants of IL-12Fc that do not interact with FcRn. The binding of FcRn to IgG Fc relies predominantly on the interaction of amino acids around positions 253, 308-311, and 435[24,25]. Of those, amino acids 310 and 435 are histidines, whose side chains become protonated at acidic pH. This increase in positive charge at the IgG:FcRn interface surface thus drives the high affinity to FcRn at low pH (<6.0)[26]. We therefore introduced amino acid substitutions at positions I253, H310, and H435 (IHH, WT configuration) in the Fc portion of the fusion molecule (Table S1). Most substitutions resulted in stable proteins which were readily purifiable as homodimers (Fig. S2A, B).

Next, we measured the affinity of different IL-12Fc variants to FcRn by surface plasmon resonance (SPR). Immobilizing human FcRn and using IL-12Fc variants in solution, we observed that the I253N H435Q combination of substitutions (NHQ) had the most profound effect on reducing FcRn affinity (Figs. 2A, B and S2C). Relative to WT (IHH), as well as previously published IAQ[27] and AAA[24] variants, NHQ IL-12Fc exhibited reduced FcRn binding, also by ELISA and sustained bioactivity on a reporter cell line and on human peripheral blood mononuclear cells (PBMCs) (Fig. S2D–F, Table S1).

We then compared retention of the WT, IAQ, AAA, and NHQ IL-12Fc variants in the brain after administration into the striatum of hFcRn Tg32 animals by CED. After 6 h we detected at least 30% higher levels of IL-12 in the brains of mice given IL-12Fc NHQ, compared to the IAQ and AAA variants and WT IL-12Fc (Fig. 2C); this difference remained detectable at 24 h post administration but grew smaller with time (Fig. S2G). If the observed increase in residual brain levels of IL-12Fc NHQ was due to abrogation of FcRn-mediated efflux, occupying the IgG binding site of FcRn in endothelial cells should have a similar effect on the WT variant. Indeed, prior intravenous administration of a modified human IgG1 Fc fragment with a pH-independent high affinity for FcRn (MST-HN Fc)[28,29] also led to over 50% higher residual brain levels of the IL-12Fc WT variant 6 h after CED, essentially reaching the same level as IL-12Fc NHQ and abolishing the significant difference observed without MST-HN Fc (Fig. 2D). In contrast, local co-administration of MST-HN Fc did not increase residual brain levels of the WT variant (Fig. S2H). FcRn-silenced IL-12Fc NHQ also showed better brain retention and lower systemic load over 7 days post-injection, in particular compared to rIL-12: both effects were most prominent up to 3 days post-injection (Figs. 2E and S2I, J and Table S2), during which the total brain exposure to IL-12Fc NHQ was increased by 50% compared to rIL-12, while the overall plasma footprint was nearly 10-fold reduced (area under the curve, AUC, Fig. 2F).

Overall, the FcRn-silenced IL-12Fc NHQ variant showed increased retention in the brain and reduced systemic prevalence upon local intraparenchymal administration, likely due to reduced transcytosis at the BBB endothelium and abolished recycling in the periphery.

### FcRn-silenced IL-12Fc exhibits durable, local anti-tumor effects in glioma-bearing mice

To assess whether this confinement to the brain parenchyma translated to an improved therapeutic window in established murine models of GBM, we first generated mIL-12hFc WT and mIL-12hFc NHQ and confirmed that they possessed similar biochemical properties to their human counterparts and comparable bioactivity to murine rIL-12 (Fig. S3).

We then administered either rmIL-12, mIL-12hFc WT or mIL-12hFc NHQ i.t. to C57BL/6 WT mice (hFcRn Tg32 we deemed not suitable as mFcRn deficiency negatively impacts immune competence[30,31]) bearing well established orthotopic GL-261 luciferase-expressing tumors, on days 21 and 28 post-tumor inoculation (Fig. 3A). Immunostained tissue sections demonstrated that delivery of 5 μl of infusate into the putative tumor center on day 21 led to partial coverage of the substantial tumor mass, potentially due to outward interstitial fluid flow towards the lateral ventricle[32] (Fig. 3B). From the first i.t. administration on day 21, mice underwent daily clinical scoring, weekly bioluminescent imaging (BLI) to measure tumor progression, and regular blood sampling until day 48. All IL-12 treatments significantly prolonged animal survival, but only mIL-12hFc WT and NHQ triggered durable responses in 15% and 30% of animals, respectively, which lasted for over 3 months after cessation of treatment (Fig. 3C, D). Upon i.t. administration, we observed substantial differences in IL-12 and IFNγ blood peak levels (Fig. 3E, F, H, I) and cumulative exposure (Fig. 3G, J) during treatment: mice treated with rmIL-12 reached a mean IL-12 peak plasma level of almost

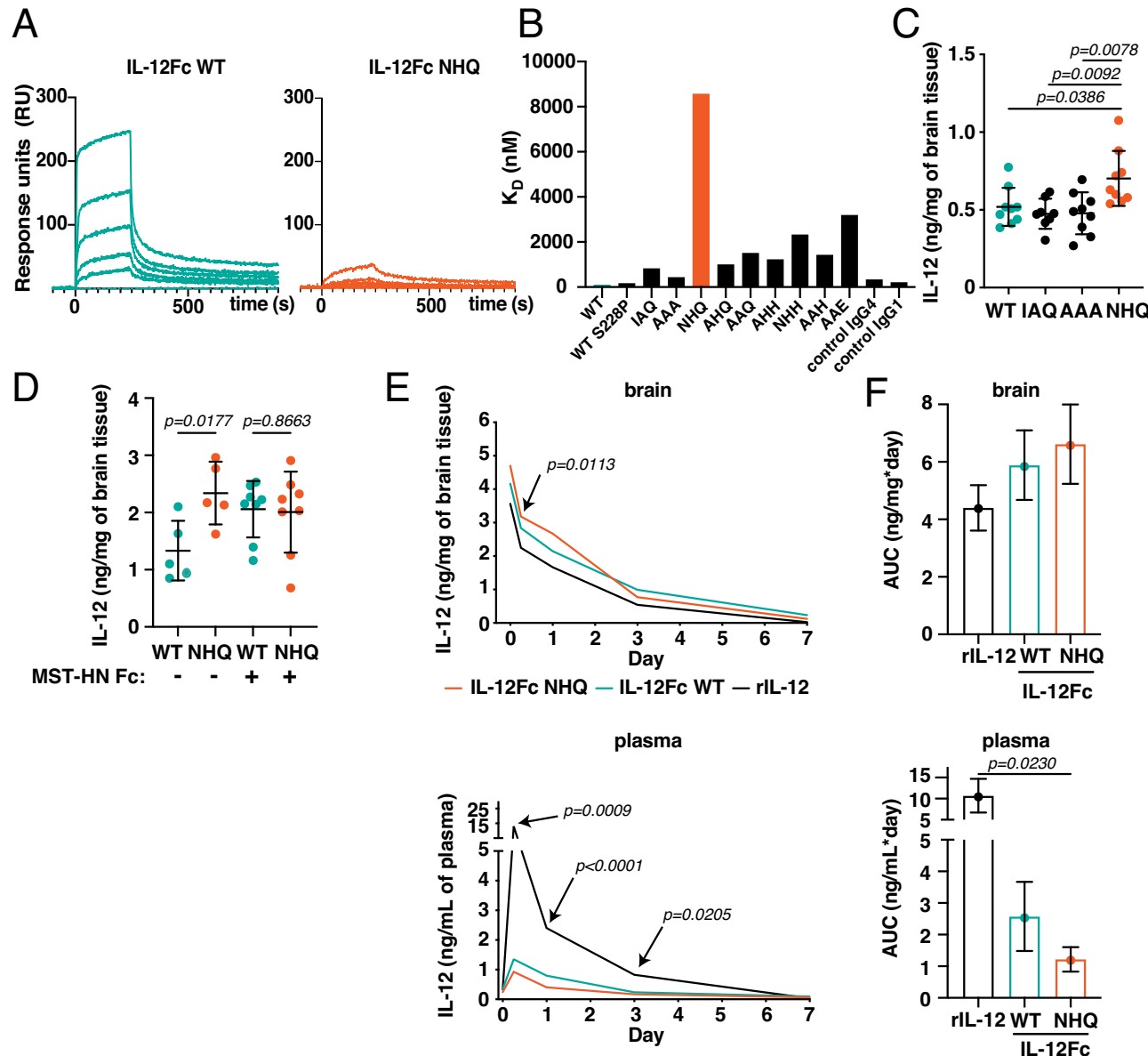

**Fig. 2 | Silencing FcRn binding increases tissue retention and reduces peripheral exposure of IL-12Fc. A** Binding of modified IL-12Fc to FcRn at pH = 6.0 was determined by surface plasmon resonance (SPR) with hFcRn in the immobilized phase. Signal response (RU) over time for IL-12Fc WT (left) or IL-12Fc NHQ (right). Curves relate to concentrations (nM, top to bottom): IL-12Fc WT: 365 – 122 – 41 – 14 – 5; IL-12Fc NHQ: 3645 – 1215 – 405 – 135 – 45. **B** Dissociation constant KD (nM) of IL-12Fc WT (IHH) and modified variants as well as control antibodies, calculated from SPR equilibrium analysis (Fig. S2C). **C** IL-12 levels in the ipsilateral hemisphere 6 h after intraparenchymal administration of 1 µg of the indicated IL-12Fc variants via CED into hFcRn Tg32 mice. Mean ± SD. Outlier removal based on Grubbs' test. One-way ANOVA with Tukey's multiple comparison test. IL-12Fc WT: $n = 9$, IL-12Fc IAQ: $n = 8$, IL-12Fc AAA: $n = 9$, IL-12Fc NHQ: $n = 9$ animals. **D** IL-12 levels in the ipsilateral hemisphere 6 h after intraparenchymal administration of 1 µg of the indicated IL-12Fc variants via CED into hFcRn Tg32 mice treated intravenously (i.v.) with 100 µg of the FcRn inhibitor MST-HN Fc.

Mean ± SD. Unpaired two-tailed $t$-test of groups IL-12Fc NHQ to IL-12Fc WT group with ($p = 0.0177$) and without FcRn inhibitor (non-significant, $p = 0.8663$). IL-12Fc WT: $n = 5$, IL-12Fc NHQ: $n = 5$, IL-12Fc WT + MST-HN Fc: $n = 8$, IL-12Fc NHQ + MST-HN Fc: $n = 8$ animals. **E** Extended pharmacokinetic (PK) analysis. Indicated IL-12 variants were administered to hFcRn Tg32 mice ($n = 7$–12 animals/timepoint/group; detailed group sizes in Table S2) and analyzed as in **C**). Mean concentration of IL-12 in brain (top) and plasma (bottom) across the next 7 days. Outlier removal based on Grubbs' test. Separate curves with individual data points in Fig. S2I, J. One-way ANOVA with Dunnett's test IL-12Fc NHQ vs IL-12Fc WT and rIL-12. Statistically significant difference only to rIL-12. IL-12Fc WT: green; IL-12Fc NHQ: orange; rIL-12: black. **F** Overall exposure (area under curve, AUC) analysis for brain (top) and plasma (bottom) in **E**) until day 3. Mean ± SEM. Unpaired two-tailed $t$-test IL-12Fc NHQ vs IL-12Fc WT and rIL-12 corrected for multiple testing using Bonferroni correction. Source data are provided as a Source Data file.

10 ng/mL; while peak levels were 21× lower in the mIL-12hFc WT (0.46 ng/mL) cohort and more than 60× lower in mIL-12hFc NHQ (0.16 ng/mL) treated mice (Fig. 3E, F). Systemic exposure to rIL-12 induced a mean IFNγ peak of 2.1 ng/mL, with a 4.4-fold and 12.4-fold reduction in mIL-12hFc WT and mIL-12hFc NHQ cohorts, respectively (Fig. 3H, I). To reach a comparably low peripheral IFNγ exposure would require at least a 10-fold reduction of the effective i.t. rIL-12

dose (Fig. S4), which would almost certainly impact therapeutic efficacy.

When we analyzed the tumor-bearing hemisphere of treated animals 1 week after the first i.t. administration of IL-12 compounds, flow cytometric analysis indicated increased infiltration of T cells—particularly of CD4+ T cells—across all IL-12-treated groups; alongside, effector CD4+ T cells, regulatory T cells (Treg) and CD8+ T cells all

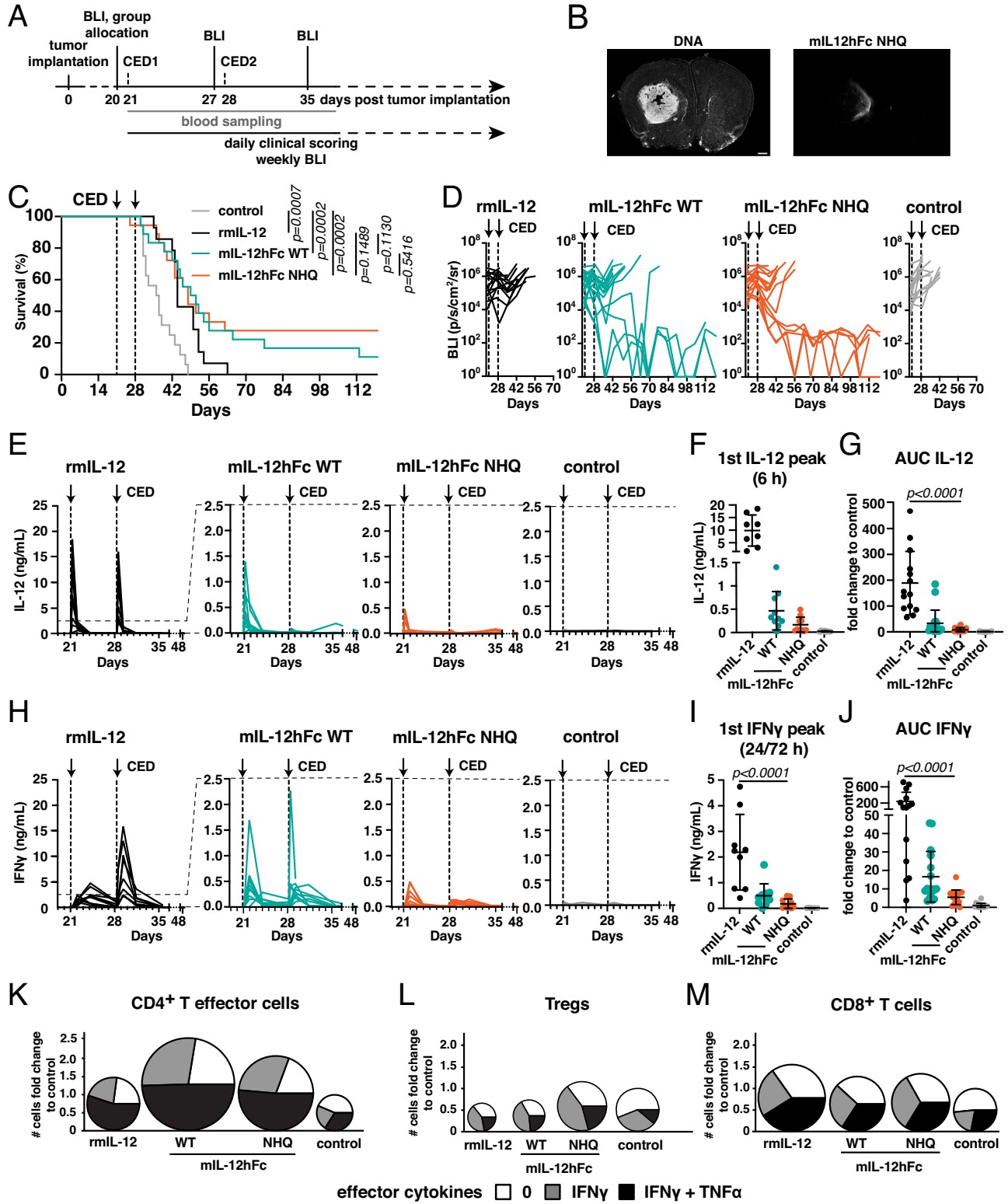

**Figure (A–M)**

showed a higher frequency of IFNγ and TNFα expression after IL-12 treatment (Figs. 3K–M and S5), indicative of increased effector functions[33] and of heightened, IFNγ-induced Treg fragility[34], as described previously during IL-12 therapy of flank tumors[35]. Overall, local treatment of established gliomas with all IL-12 variants was associated with increased survival and pro-inflammatory lymphocyte polarization compared to untreated controls, with FcRn-silenced mIL-12hFc additionally exhibiting the highest frequency of durable clinical responses and lowest systemic exposure to IL-12.

**FcRn-silenced IL-12Fc is well-tolerated in IL-12-sensitive mice**

While peripheral IFNγ is a sensitive biomarker for systemic IL-12 exposure due to leakage, mouse strains differ in their susceptibility to IL-12 toxicity[10,11]. The C57BL/6 strain, in which most of today's GBM models have been established, is comparably tolerant[36], and accordingly, we did not observe any overt clinical toxicity during the above studies. In contrast, in the C3H/HeJ mouse strain, rmIL-12 elicited similar adverse reactions as in humans[10]. To better understand the potential side effects of FcRn-silenced IL-12Fc, we compared tumor-

**Fig. 3 | Increased therapeutic index of IL-12Fc NHQ in murine GBM. A** GL-261:luc cells were injected into the right striatum of WT animals. At day 20, animals were allocated to treatment cohorts with comparable tumor load as assessed by in vivo bioluminescent imaging (BLI), and 1 μg of IL-12 variants was administered intratumoral (i.t.) via CED the next day. Blood samples were taken on day 21 before CED and 6 h after, then on day 22, 24, and 28 (before CED and 6 h after), as well as on days 29, 31, 35, and 49. **B** Representative frontal overview of a treated GL-261:luc tumor on day 21. Animals were sacrificed 5 min after i.t. administration of mIL-12hFc NHQ as described in **A**. Nuclear stain (left) and mIL-12hFc NHQ infusate (right). Scale bar = 0.5 mm. **C** Kaplan–Meyer curves of mice described in **A**. Log-rank (Mantel−Cox) test. Data pooled from three independent experiments. Control (gray): n = 16, rmIL-12 (black): n = 14, mIL-12hFc WT (green): n = 18, mIL-12hFc NHQ (orange): n = 18 animals. **D** BLI traces (average radiance) of individual animals described in **A**, treated with rmIL-12, mIL-12hFc WT, mIL-12hFc NHQ, or buffer control. Dashed vertical lines indicate CED. **E–J** Systemic cytokine levels of individual animals treated as described in **A**, **C**: IL-12 (**E−G**) and IFNγ (**H−J**) assessed by multiplex bead assay. **E, H** Plotted plasma concentration by days post-CED. Control: n = 16, rmIL-12: n = 14, mIL-12hFc WT: n = 18, mIL-12hFc NHQ: n = 18 animals. Data pooled from **C**, **D**. Dashed horizontal lines depict the change in the axis scale between the plots for rmIL-12 and other experimental cohorts. Peak values of plasma cytokine levels (**F**) IL-12 (6 h) and (**I**) IFNγ (24 h; 72 h for rmIL-12 group). Mean ± SD. Control: n = 10, mIL-12hFc WT: n = 10, mIL-12hFc NHQ: n = 10; rmIL-12: n = 8 animals. One-way ANOVA with Dunnett's test mIL-12hFc NHQ to mIL-12hFc WT, rmIL-12, and control groups. Overall exposure (AUC) of **G** IL-12 or **J** IFNγ, expressed as fold change to control group. Mean ± SD. One-way ANOVA with Dunnett's test mIL-12hFc NHQ to mIL-12hFc WT, rmIL-12, and control groups. Data pooled from two (**E, F, H, I**) or three (**G, J**) independent experiments from **C**. TNFα and IFNγ positive effector CD4+ T (**K**), Tregs (**L**), and CD8+ T cells (**M**) in the tumor-bearing hemisphere on day 28, as analyzed by flow cytometry. Diameters of charts correspond to the fold change in cell counts compared to the control group. Gating strategy depicted in Fig. S5A. Data pooled from two independent experiments. Source data are provided as a Source Data file.

naïve C3H/HeJ mice receiving two intraparenchymal doses of either 1 μg or 5 μg mIL-12hFc WT, mIL-12hFc NHQ, or rmIL-12, 7 days apart (Fig. S6A). Our initial focus was systemic toxicity. At both doses, mIL-12hFc WT and rmIL-12 led to weight loss, substantial increases in spleen size, and increased levels of plasma IL-12 and IFNγ, as well as two markers of liver damage, CCL-2 and alanine transaminase (ALT) (Fig. S6B–J). Mice treated with mIL-12hFc NHQ showed significantly less peripheral toxicity according to these soluble measures, with more than 2-fold lower ALT and CCL-2 levels and more than 10-fold lower IFNγ levels at the 5 μg dose.

To extend our study to organ toxicity, we compared rmIL-12 and FcRn-silenced IL-12Fc in more depth: body and spleen weight confirmed the initial findings of overt toxicity of 5 and 1 μg of rmIL-12, compared to well-tolerated mIL-12hFc NHQ (Fig. 4A, B). As previously observed, local administration of rmIL-12 also led to higher plasma levels of IL-12 (Fig. 4C, D) and the associated IFNγ and CCL-2 (Fig. 4E–H) at both doses. Extensive pathological examination revealed substantial alveolar damage, edema, and pulmonary hemorrhages in males that received 5 μg of IL-12. This was not observed in female mice, likely because they had to be sacrificed before the planned endpoint due to severe weight loss, and lung toxicity was not yet fully developed. There were also signs of cardiac degeneration in mice given rmIL-12, but not in those injected with mIL-12hFc NHQ (Fig. 4I).

According to histological examination of brain morphology, focusing on neuronal damage and signs for apoptosis (H&E staining, alongside NeuN and cleaved caspase-3 immunolabeling), tissue damage around the putative target coordinates appeared primarily related to the invasive local delivery and less to rmIL-12 or mIL-12hFc NHQ. Microgliosis (Iba-1) and astrogliosis (GFAP) were moderate and comparable between test compounds, and showed a weak dose dependency (Fig. 4I). Overall, the confinement of IL-12 bioactivity to the brain parenchyma in the case of mIL-12hFc NHQ did not appear to trigger overt neurotoxicity, and protected mice susceptible to systemic IL-12/IFNγ toxicity from the severe weight loss and tissue damage associated with rmIL-12 treatment.

**Local FcRn-silenced IL-12Fc increases the efficacy of chemotherapy despite peripheral lymphopenia**

Surgical removal of the tumor followed by alkylating chemotherapy with temozolomide (TMZ) in combination with radiotherapy represents the standard of care (SoC) for most patients with GBM[1], yet despite these treatments, their prognosis remains grim. We reasoned that local inflammatory reprogramming of the immunosuppressive microenvironment via IL-12, combined with antigen release by chemotherapy[37] might act synergistically. We thus tested treatment of GL-261 tumors with systemic TMZ in combination with i.t. mIL-12hFc NHQ at d21 and d28 (Fig. 5A). We observed a significant prolongation of survival with both monotherapies, which was further increased in

the group treated with both mIL-12hFc NHQ and TMZ (Figs. 5B and S7A). Of note, this occurred despite the expected TMZ-induced peripheral lymphopenia[37], which resulted in substantially reduced numbers of T cells, NK cells, and B cells in the blood in both TMZ-treated groups (Fig. 5C). Strikingly, and in contrast to this notion, we observed a local increase in CD4 and CD8 T cell numbers (Figs. 5D and S7B, C) as well as microglia and to a lesser degree, myeloid cells (Fig. S7D) in the cohorts treated with mIL-12hFc NHQ. Numbers of antigen-experienced PD-1+ T cells[38] (Fig. S7E) increased as well as overall expression of CD69, a marker of tissue-resident T cells[38] (Fig. S7F). Interestingly, over 70% of CD8+ T cells in the tumors co-expressed CD69 and PD-1, a phenotype of brain-resident memory T cells[39] (Fig. 5E). Moreover, IL-12 treatment triggered CD4 effector cell proliferation, as depicted by Ki67 expression (Fig. 5F) and microglial activation as demonstrated by enhanced MHCII expression (Fig. 5G).

The clear survival benefit in the context of severe systemic lymphopenia and apparent local proliferation of effector T cells led us to ask whether peripheral T cell influx was an expendable feature for locally confined FcRn-silenced IL-12Fc and if this would also apply to IL-12 with its higher systemic load. Staying with the GL-261 model, we used normalized amounts of rmIL-12 or mIL-12hFc NHQ (compare Fig. S4) in the presence or absence of a very-late-antigen-4 (VLA-4) blocking antibody, which prevents VLA-4 integrin-mediated lymphocyte migration and infiltration into tissues−including brain tumors[40,41]. Despite systemic VLA-4 blockade, both rmIL-12 and mIL-12hFc NHQ triggered comparable increases in survival over controls (Fig. S8A, B). In addition, while IL-12 also triggers CD8 influx, VLA-4 blockade inhibited CNS influx of CD4 rather than CD8 T cells (Fig. S8C–E).

**FcRn-silenced IL-12Fc synergizes with radiotherapy in immunotherapy-resistant mouse GBM**

We next sought to test the efficacy of local therapy with CNS-contained mIL-12hFc NHQ in a more challenging and aggressive GBM model. Orthotopic SB28 tumors are derived from a genetically engineered mouse model, progress rapidly, and harbor few genetic mutations; their low immunogenicity results in resistance to anti-PD-1 and anti-CTLA-4 ICI[42]. Focal radiotherapy may at least partially overcome this unresponsiveness to immunotherapy[43]. Adjusting our treatment scheme to the fast progression of this tumor, we combined local radiotherapy (RT) at days 7 and 9 with i.t. mIL-12hFc NHQ administration at days 5 and 12 (Fig. 5H). Compared to day 21 GL-261 lesions, day 5 SB28 tumors were smaller and CED resulted in a complete coverage of the tumor mass (Fig. 5I). All treatments increased survival, but the most considerable increase was seen in mice treated with mIL-12hFc NHQ + RT, which led to up to 90% of animals initially responding and up to 30% exhibiting durable and complete remissions as assessed by three consecutive BLI readings (Figs. 5J and S9A). These responding animals were rechallenged at day 69 post-initial-tumor-cell-

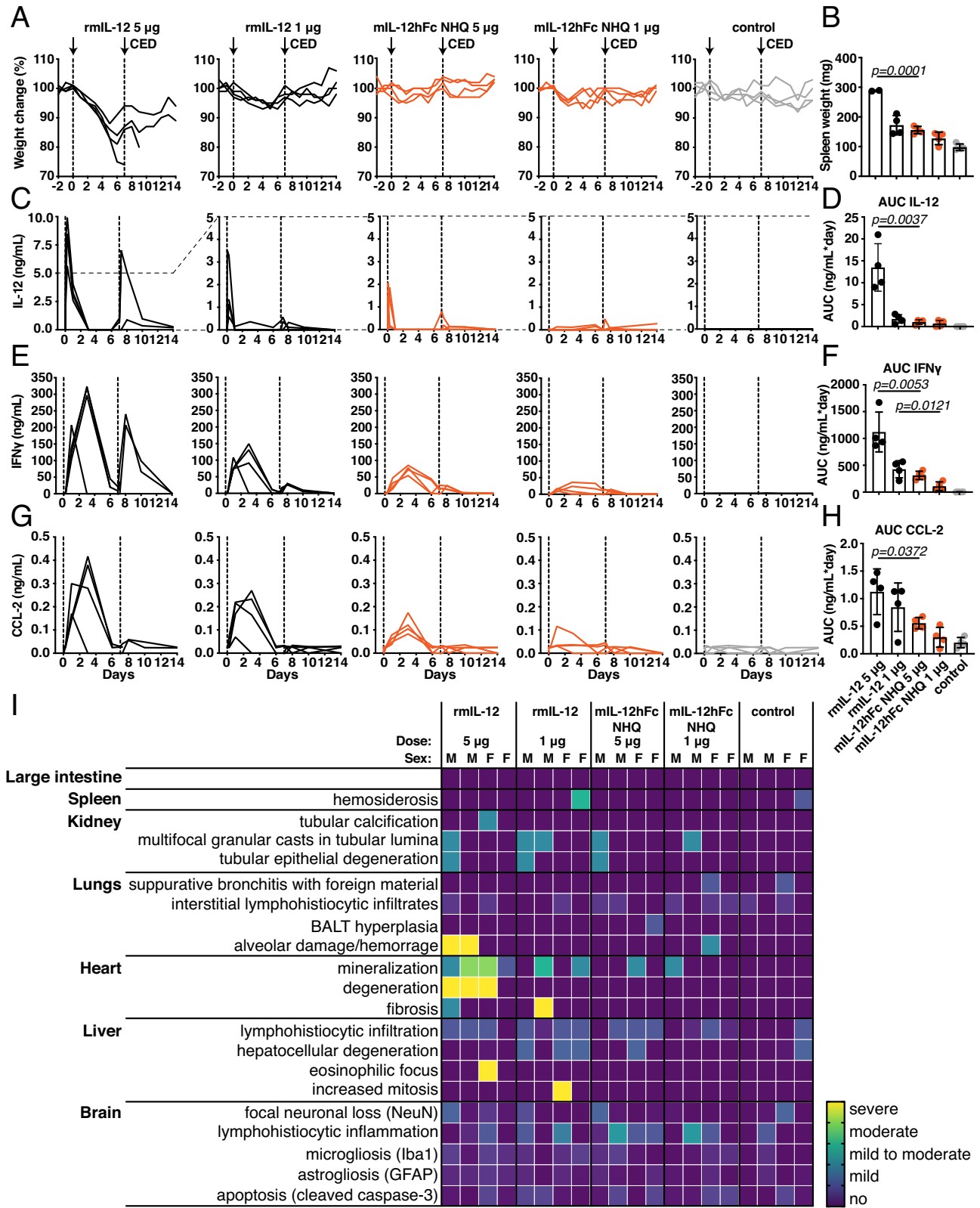

**Fig. 4 | C3H/HeJ IFNγ-sensitive mice tolerate high intraparenchymal doses of mIL-12hFc NHQ without systemic toxicity.** mIL-12hFc NHQ or rmIL-12 were intraparenchymally administered via CED at two dose levels (1 or 5 μg) on days 0 and 7. **A** Weight of individual animals in each cohort, normalized to starting weight. **B** Spleen weight. Animals that reached withdrawal criteria before the planned endpoint are not included. Unpaired two-tailed *t*-test of rmIL-12 vs mIL-12hFc NHQ groups at each dose level. Mean ± SD. Control: *n* = 4, rmIL-12 5 μg: *n* = 2, rmIL-12 1 μg: *n* = 4, mIL-12hFc NHQ 5 μg: *n* = 4, mIL-12hFc NHQ 5 μg: *n* = 4 animals. Systemic IL-12 (**C**, **D**), IFNγ (**E**, **F**), and CCL-2 (**G**, **H**) concentrations of individual animals

assessed by multiplex bead assay in plasma. Dotted vertical lines indicate CED. Dashed horizontal lines depict the change in the axis scale between the plots for rmIL-12 5 μg and other experimental cohorts. Overall exposure (AUC) was analyzed by an unpaired two-tailed *t*-test of rmIL-12 vs mIL-12hFc NHQ groups at each dose level. Mean ± SD. Control: *n* = 4, rmIL-12 5 μg: *n* = 4, rmIL-12 1 μg: *n* = 4, mIL-12hFc NHQ 5 μg: *n* = 4, mIL-12hFc NHQ 5 μg: *n* = 4 animals. **I** Histopathological scores of animals in **A**, **B** as assessed by a board-certified veterinarian pathologist and normalized to the average of the control group. BALT bronchus-associated lymphoid tissue. Source data are provided as a Source Data file.

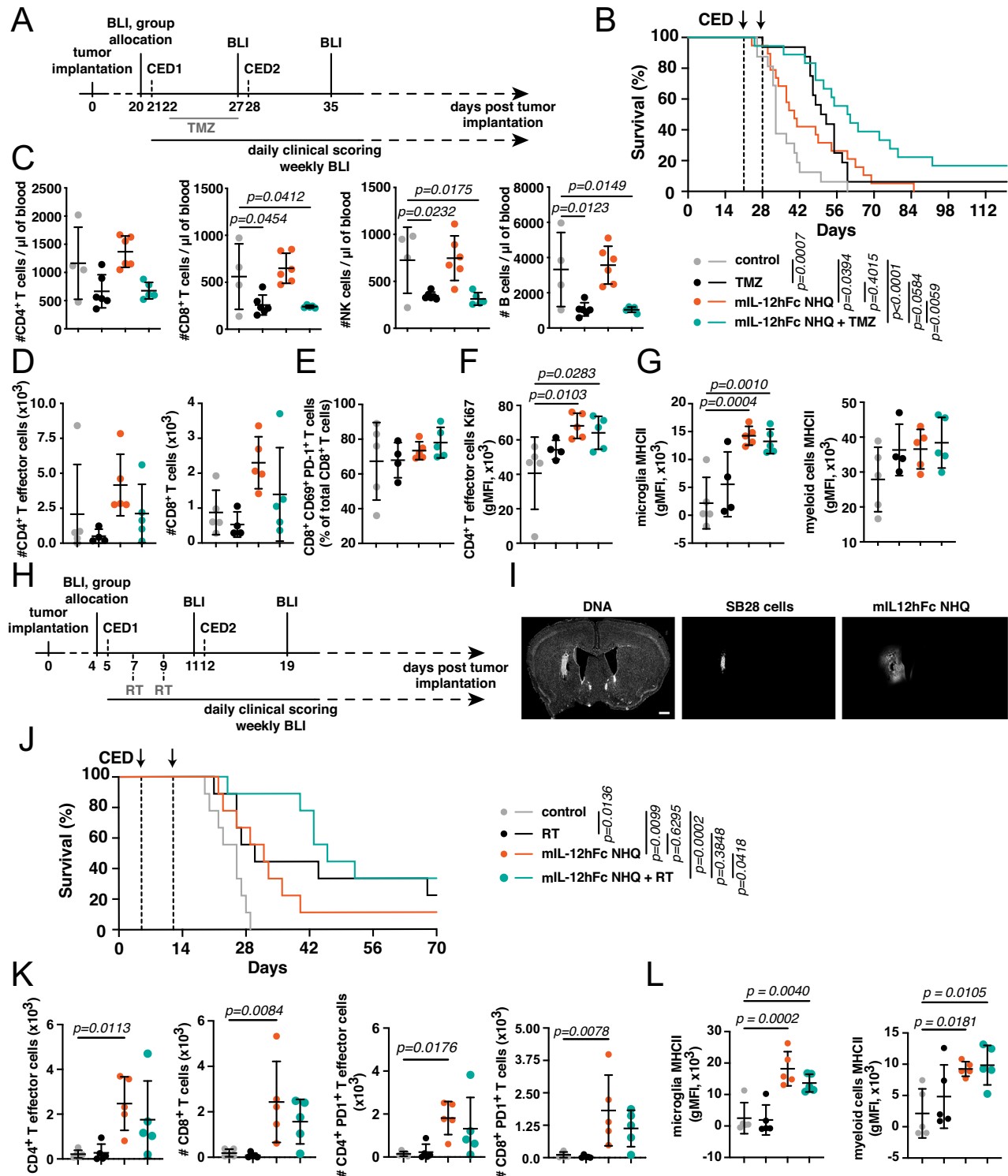

inoculation, revealing protective memory formation in half of the mIL-12hFc-treated mice (Fig. S9B). In line with previous analysis (Figs. 3K–M, S5), mIL-12hFc NHQ monotherapy triggered a significant increase in effector CD4+ and CD8+ T cells with an increased PD-1+ fraction and Tregs (Figs. 5K and S9C, D). RT did not substantially increase the frequency of lymphocytes and led to a mild reduction in the numbers of microglia and myeloid infiltrates. Increased MHCII expression on both microglia and myeloid infiltrates in the combination group and upon monotherapy with FcRn-silenced mIL-12hFc suggested inflammatory repolarization of the TME beyond lymphocytes and an increase in antigen presentation (Figs. 5L and S9E).

Overall, these data indicate that local mIL-12hFc NHQ immunotherapy synergizes with radio and chemotherapy, the current SoC in GBM, is effective also in a rapidly growing, treatment-resistant GBM model, and does not per se rely on peripheral effector T cell influx.

## FcRn-silenced IL-12Fc exhibits reduced BBB transcytosis and induces inflammatory reprogramming in patient-derived GBM explants

To understand whether the therapeutic potential of FcRn-silenced IL-12Fc would likely translate to the human setting, we employed a human BBB model consisting of FcRn-positive induced pluripotent stem cell

**Fig. 5 | FcRn-silenced IL-12Fc synergizes with chemotherapy and RT, even in models of aggressive treatment-resistant glioma. A–G** GL-261 brain tumor-bearing C57BL/6 mice were treated with mIL-12hFc NHQ as in Fig. 3 and with five daily doses of temozolomide (TMZ) between days 23 and 27, either alone or in combination. **B** Kaplan–Meyer curves for the experiment described in **A** treated with buffer control, TMZ, mIL-12hFc NHQ, or mIL-12hFc NHQ + TMZ. Log-rank (Mantel–Cox) test. Data pooled from three independent experiments. Control (gray): $n = 16$, TMZ (black): $n = 16$, mIL-12hFc NHQ (orange): $n = 19$, mIL-12hFc NHQ + TMZ (green): $n = 18$ animals. **C–G**: same legend as in **B**. **C** Flow cytometric quantification of blood CD4$^+$ T cells, CD8$^+$ T cells, NK cells, and B cells on day 28. Mean ± SD. One-way ANOVA with Dunnett's test vs control. Representative data from one of the experiments contributing to **B. D–F** Tumor-infiltrating lymphocytes (TILs), analyzed by flow cytometry at day 32, 4 days after the second CED. Gating strategy depicted in Fig. S7B. One-way ANOVA with Dunnett's test to the control group. Mean ± SD. **D** Total number of CD4$^+$ T effector cells and CD8$^+$ T cells, as well as **E** percentage of CD69$^+$ PD-1$^+$ among total CD8$^+$ T cells, **F** gMFI of Ki67 on CD4$^+$ T effector cells, **G** gMFI of MHCII on microglia and myeloid infiltrates.

**H–L** SB28 cells were injected into the right striatum of WT animals. On day 4, animals were allocated to treatment cohorts with comparable tumor load as assessed by BLI, and 1 µg of mIL-12hFc NHQ was administered intratumorally by CED. Mice received two doses of 10 Gy radiation (RT) targeted to the tumor location on day 7 and day 9. **I** Frontal overview over treated SB28 tumor (day 5 post-inoculation), 5 min after i.t. administration of mIL-12hFc NHQ via CED. Nuclear stain (left), SB28 tumor (middle), and mIL-12hFc NHQ shown (right). Scale bar: 0.5 mm. **J** Kaplan–Meyer curves of cohorts described in **H** treated with buffer control, RT, mIL-12hFc NHQ, or mIL-12hFc NHQ + RT. Log-rank (Mantel–Cox) test compared to controls. Data pooled from three independent experiments. Control (gray): $n = 9$, RT (black): $n = 9$, mIL-12hFc NHQ (orange): $n = 9$, mIL-12hFc NHQ + RT (green): $n = 9$ animals. **K, L** TILs were analyzed by flow cytometry at day 16, 4 days after the second CED. One-way ANOVA with Dunnett's test vs control. Gating as shown in Fig. S9C. Same legend as in (**J**). Total number of **K** CD4$^+$ T effector cells, CD8$^+$ T cells, CD4$^+$ PD-1$^+$ and CD8$^+$ PD-1$^+$ T cells as well as **L** gMFI of MHCII on microglia and myeloid infiltrates in the tumor-bearing hemisphere. Mean ± SD. Source data are provided as a Source Data file.

(iPSC)-derived brain microvascular endothelial cells (BMECs), which has proven well-suited to study the transport of biologics[44] (Figs. 6A and S10A). We applied IL-12 variants to the basolateral–putatively brain-parenchyma-facing compartment–and measured IL-12 levels 24 h later in the apical reservoir. Consistent with our observations in hFcRn Tg32 mice, we observed a higher apparent transport rate for rIL-12 compared to IL-12Fc variants (Fig. 6B). In line with the finding that inhibition of FcRn in the endothelium rather than the brain parenchyma phenocopies the NHQ modification, FcRn-silenced IL-12Fc showed further reduced basolateral to apical transport rates (Fig. 6B). Interestingly, a similar finding was obtained in the apical-to-basolateral direction (Fig. S10B).

Given the apparent dispensability of T cell influx for survival benefit in the TMZ-treated mouse model (Fig. 5B, C), we speculated that IL-12-mediated reactivation of TME-resident bona fide tumor-reactive T cells might be an important effector mechanism of i.t. IL-12 therapy. To evaluate this possibility in a translationally relevant setting, we performed two series of perfusion bioreactor assays using resected GBM tissue from newly diagnosed patients (Fig. 6C, Table S3). This approach allows the preservation of tissue integrity and thus the TME[45,46]. Tumor pieces derived from the viable center[45,46] were trimmed and placed in parallel in bioreactors containing 100 ng/mL IL-12Fc NHQ or control medium only (Fig. 6C). We then measured the abundance of 92 soluble proteins, including cytokines, chemokines and soluble membrane proteins, in the medium of each bioreactor using proximity extension assay technology and empirically adjusted sample weights based on the assay variability of the pooled patients (Fig. S11). Next to the exogenous IL-12Fc NHQ (detected as IL-12 by the assay) several analytes were significantly enriched upon treatment (Fig. 6D). Most strikingly, there was a substantial upregulation of IFNγ and its hallmark pathway analytes (Fig. 6E), with individual effector molecules including Fas ligand (FASLG), granzyme B (GZMB) and H (GZMH) showing marked increases (Fig. 6F). In addition, chemokines stimulating lymphocyte infiltration including CXCL9, CXCL10 and CXCL11 as well as OX40 ligand (TNFSF4) were increased; while Arginase 1 (ARG-1) and ICOS ligand (ICOSLG)–two analytes associated with an immunosuppressive TME–were reduced (Fig. 6F). These patterns of pro-inflammatory activation and upregulation of chemotactic cues were remarkably conserved across most patients (Fig. 6F). Overall, in a human BBB transport ex vivo assay IL-12Fc NHQ appeared to be better-retained compared to rIL-12 or IL-12Fc WT, and was bioactive in patient-derived GBM TME, leading to inflammatory polarization independent of influx of peripheral lymphocytes.

## Discussion

Local, i.t. application of strong immunomodulators can circumvent dosing limitations and promises to potentiate the effect of therapeutic proteins by increasing their abundance in the TME[47]. Unfortunately, many cytokines for cancer immunotherapy, such as IL-2 and IL-12, demonstrate poor intrinsic retention upon i.t. administration largely due to their small size[13,14,48,49]. Previous studies have successfully tested the potential of conjugating immune molecules to collagen-binding proteins or of increasing construct size as a means of promoting retention in flank tumors in mice[14,48]. Here, we achieved a similar size increase of IL-12 by incorporating it into a bivalent Fc-fusion format[15], which translated to higher retention in the brain parenchyma upon local administration, and more long-term responses in glioma-bearing mice after treatment.

The Fc that we chose to use was from IgG4, as this class is relatively inert with regard to complement binding and Fc gamma receptor engagement, but it is readily bound by FcRn[16]. Surprisingly, we uncovered that FcRn is expressed not only in the healthy brain but at even higher levels in the GBM brain–not only in the vasculature, but potentially also in the tumor parenchyma, where it could play a pathophysiological role[50]. In the local treatment setting, FcRn supports increased systemic exposure to IL-12Fc through its role in transcytosis and recycling of endocytosed IgG[8,24]. Continuous plasma accumulation of IL-12Fc leaked from the brain tumor, which was abolished in the absence of FcRn:Fc interactions, prominently revealed this.

To exploit the advantageous retention of the IL-12Fc fusion protein in the brain, while circumventing FcRn export to the blood, we mutated the FcRn interaction interface of IgG Fc. Next to substitutions derived from traditional alanine screens[24,51], we also tested the effects of using uncharged amino acids with similar side chain length and flexibility, eventually selecting the NHQ substitution variant. In vivo, the NHQ mutant showed higher brain retention early after intraparenchymal administration and reduced plasma prevalence. Here, we now studied brain-to-blood transcytosis in a human FcRn transgenic mouse background with human cytokine Fc fusions instead of antibodies: the hFcRn Tg32 mouse line not only recapitulates human FcRn expression at the BBB[18], but has also been extensively validated for pharmacokinetic (PK) analyses of antibodies, including PK prediction of human FcRn affinity-modified antibodies[19,52]. As observed by Cooper and colleagues for human antibodies injected into rats[53], we detected increased retention of the protein in the brain after reducing its affinity for FcRn. Next to intermittent delivery, FcRn silencing may be particularly relevant for containing the activity of continuously infused or in-situ-expressed biologics. Our study now adds evidence in a humanized in vivo setting for FcRn-mediated transcytosis from brain to blood after local application.

Both our in vivo data as well as our results in an ex vivo BBB model align with the swift leakage of rIL-12 from the brain tumor parenchyma as reported by Chiocca and colleagues during local IL-12 gene therapy[13], and demonstrate a significant reduction of such transport in

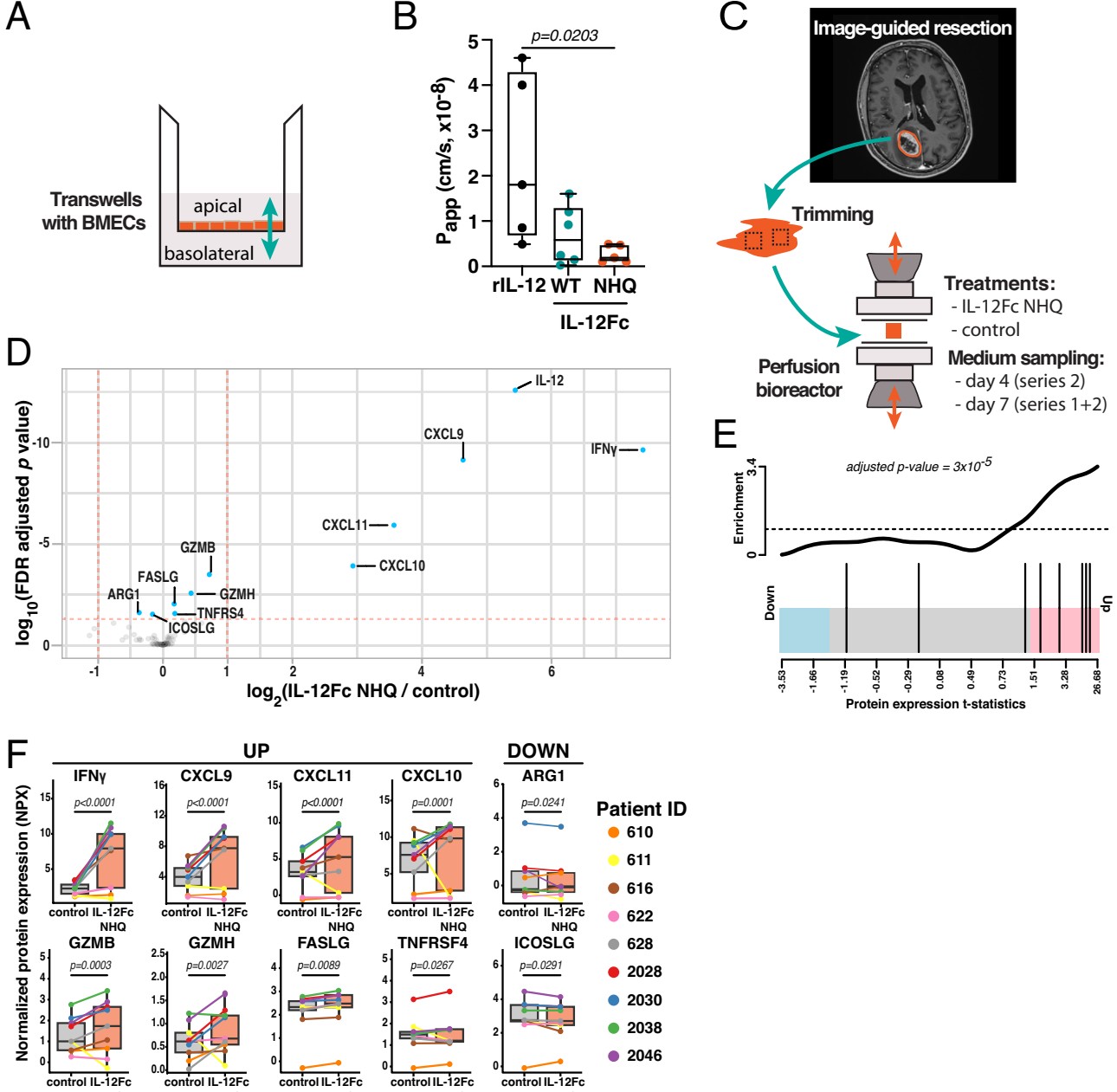

**Fig. 6 | Proinflammatory conditioning of FcRn-silenced IL-12Fc-treated patient-derived GBM explants with intact tumor microenvironment. A** Scheme of an in vitro blood-brain barrier (BBB) model formed by a brain microvascular endothelial cell (BMEC) monolayer in a transwell insert, derived from human induced pluripotent stem cells (iPSCs). **B** Transport rates of IL-12 variants across a BMEC monolayer in basolateral (brain) to apical (blood) direction. One-way ANOVA with Dunnett's test to the IL-12Fc NHQ group. Outlier removal based on Grubbs' test. rIL-12: $n = 6$, IL-12Fc WT: $n = 6$, IL-12Fc NHQ: $n = 5$ experiment repetitions. Whiskers represent the minimum to maximum, middle line represents the mean. **C** Tumor explants from nine patients with newly diagnosed GBM were trimmed, divided into parallel perfusion bioreactor cultures, and treated with 100 ng/mL IL-12Fc NHQ or control medium for up to 7 days. Supernatants were analyzed using multiplex Proximity Extension Assay (PEA Olink Target 96 IO). **D** Differentially expressed proteins in supernatants of bioreactors treated with IL-12Fc NHQ compared to control medium. Values with a false discovery rate (FDR) < 0.05 (horizontal red line) are considered significant. The vertical red line marks a fold change of 2. **E** Barcode plot of the hallmark IFNγ gene set. The vertical bars in the lower part of the plot mark the $t$-statistic of those genes which are in the gene set against the background expression of the remaining genes in the panel. **F** Sign plots of normalized protein expression (NPX) at day 7, comparing IL-12Fc NHQ vs control-treated bioreactors. The middle line marks the sample median, the hinges correspond to the first and third quartiles, and the whiskers extend up to a further ±1.5 times the length of said inter-quartile range. All protein level $p$ values are FDR-adjusted and are derived from moderated $t$-statistics. $n = 9$ patients/group. Source data are provided as a Source Data file.

the case of the FcRn-silenced IL-12Fc variant NHQ. The intermediate transport rates that we detected for the IL-12Fc WT variant suggest that both larger size and abolished FcRn affinity may contribute to this lower export in a human setting. The finding that FcRn expression is upregulated in GBM and contributes to the efflux of IL-12Fc is therefore also relevant for local administration of other biologics, such as antibodies used in ICI therapy[7,54], in particular for i.t. treatment with a CD40 agonistic antibody (NCT03389802) of patients with GBM.

In the clinical setting, local delivery of therapeutics can be achieved by freehand injection into the residual tumor tissue or the

cavity walls after resection[7]. An option for inoperable tumors or administration prior to resection is catheter-based delivery methods, for example, i.t. delivery via CED as used in the here presented pre-clinical study[17] and in various clinical trials[54,55]. When treating established GL-261 tumors via CED, we observed increased survival with all IL-12 versions tested; yet only in IL-12Fc-treated cohorts were durable responses apparent, especially with the FcRn-silenced NHQ variant, which could be indicative of prolonged i.t. activity. Next to higher CNS retention, the IL-12Fc NHQ variant has a substantially reduced risk for systemic, dose-limiting toxicity: in the IL-12-sensitive C3H/HeJ mouse strain, unmodified rmIL-12 as well as the mIL-12hFc WT variant elicited substantial toxicity. This was likely due to high systemic IFNγ levels after leakage and is in line with dose-limiting systemic rather than neurological toxicity during local IL-12 gene therapy in human GBM clinical testing[13]. On the other hand, local administration of up to 5 μg of IL-12Fc NHQ resulted in microgliosis and astrogliosis, but not in overt neurotoxicity or autoimmune attack.

Despite their different in vivo PK behavior, both human and murine IL-12Fc variants displayed a comparable bioactivity profile compared to their unmodified counterparts. Our initial observations align with previously described immune correlates in the TME upon local IL-12-treatment of murine and human GBM: increased i.t. T cells and IFNγ production have been reported upon local CNS expression in transgenic mice[56], expression in genetically engineered tumor cells[22], viral gene delivery[13,33,57] or direct local infusion[22,58]. While a more detailed characterization of the TME and further mechanistic investigation is warranted, our findings so far suggest a comparable mode of action between IL-12, IL-12Fc WT, and the IL-12Fc NHQ variant upon local GBM treatment. However, with a negligible cytokine footprint in such a setting, the latter showed little to no signs of systemic side effects.

As previously observed when testing local IL-12 GBM gene therapy[36], FcRn-silenced IL-12Fc synergized with systemically delivered TMZ, despite substantial TMZ-induced peripheral lymphodepletion. In the TME, however, the majority of intratumoral CD8+ and, to a lesser extent CD4+ T cells, expressed markers of antigen-experienced, tissue-resident cells[39] and treatment with FcRn-silenced IL-12Fc triggered proliferation of CD4+ effector T cells, a crucial population for anti-GBM efficacy[22,33,57]. Local immunity driven by reactivation of existing tumor-reactive T cells, TMZ-induced antigen release, and antigen presentation by activated microglia may directly contribute to the increased efficacy of FcRn-silenced IL-12Fc and also to an eventual broadening of the immune response. In addition, robust treatment responses, even with pharmacological inhibition of T cell CNS influx during mono-therapy by VLA-4 blocking antibodies, suggested that recruitment of peripheral effector T cells is likely not absolutely required for therapeutic efficacy.

With its high mutational load, GL-261 is more immunogenic than human GBM[42]. Notwithstanding a solid correlation of its immune infiltrate to human GBM[59], a higher number of T cells may overestimate the impact of local reactivation of i.t. T cells with the TME-confined IL-12Fc NHQ variant. Therefore the high frequency of responders in the aggressive and much less immunogenic genetically engineered ICI-resistant SB28 model is encouraging: the number of durable responses when combined with radiotherapy and the partial protection against orthotopic rechallenge in the contralateral hemisphere suggests induction of immunological memory, which is rarely seen in SB28[43,60] and to our knowledge has not been demonstrated yet in a fully orthotopic primary treatment and contralateral rechallenge setting. Indeed, the antigen-release effect of irradiation in combination with enhanced antigen presentation and priming via IL-12 may be responsible for this long-lasting effect[43]. In addition, it is tempting to speculate that senescence of SB28 GBM cells triggered directly by irradiation and exacerbated by IL-12-induced, T cell-derived IFNγ and TNFα in the TME may be responsible for the high number of initially

responding animals. Such an effect has been suggested previously for an IL-12 immunocytokine in combination with radiotherapy[61] and is conceivable here too, based on increased IFNγ and TNFα expression in tumor-infiltrating lymphocytes (TILs). In addition, in the secretome of IL-12Fc NHQ-treated patient-derived brain tumor explants, IFNγ and its hallmark gene signature members were significantly increased. Adjuvant radiotherapy is a cornerstone of the SoC for GBM patients, and the intraoperative administration of FcRn-silenced IL-12Fc to residual tumor parenchyma, as has been demonstrated for intraparenchymal ICI administration, should be feasible[7].

Overall, treated patient explants revealed a profound reversal of local immunosuppression putatively driven by M2-like glioma-associated microglia and myeloid infiltrates, as suggested by Arginase 1[62] and regulatory T cells fostered by ICOSL[63] to an inflammatory hotspot, characterized by cytotoxic mediators including FASLG, GZMB, GZMH, chemokines CXCL9, CXCL10 and CXCL11[64,65] and the T effector cell activating TNF receptor family member OX40[66]. While GBM overall is considered a weakly immune-infiltrated tumor due to its low mutational burden and anatomical location[42,67,68], the above signs of inflammatory conditioning support the notion that the viable tumor center can contain considerable numbers of potentially exhausted, yet IL-12-responsive and readily activatable populations of tumor-reactive effector CD4 and CD8 T cells[45,46]. As in an actual clinical setting, patient-to-patient heterogeneity in the response to IL-12Fc NHQ is expected and may in part be attributable to a few samples with comparably low signal intensity and high assay variability. However, a comprehensive assessment to formulate future clinical expectations will require a larger cohort.

Taken together, we confirmed the rapid leakage of IL-12 from brain-to-blood over the BBB upon i.t. application—as previously reported during clinical testing[13]—both in vivo in mice and in a human BBB model. An IL-12Fc fusion format with increased size improved tissue retention, but the Fc:FcRn interaction at the BBB endothelium drove brain-to-blood transcytosis and subsequent recycling. We have now added in vivo proof of this notion in a fully humanized setting[8]: both pharmacologic inhibition and abrogation of the FcRn:Fc interaction by amino acid substitutions further increased brain retention and contained IL-12 activity within the CNS. Quantitative differences in PK and pharmacodynamic behavior and toxicity will need to be confirmed in large animal species. Nonetheless, our data indicate that the FcRn-silenced IL-12Fc NHQ variant enables a substantially wider local dosing window concerning treatment response, dose-dependent leakage, and toxicity. The finding that both murine GBM models and patient explants with intact TME contain a critical number of IL-12-responsive effector cells that can be reactivated encourages clinical translation. Beyond brain cancer, FcRn silencing is in principle applicable to various Fc-containing biological formats. Such an approach, therefore, may also prove useful for other CNS diseases to restrict the activity of antibodies or Fc-fusion proteins to the brain parenchyma.

## Methods

### Ethics approval

All procedures described in the present study were approved by the Cantonal Veterinarian's Office of Zurich (Licenses ZH246/2015, ZH194/2019, and ZH175/2022), and all efforts were made to minimize the number of animals used and their suffering. Humane endpoint criteria applied to the tumor and toxicology studies included body weight (BW) loss >20% of peak weight, and/or clinical findings such as hunchback posture, lack of activity, seizures, or loss of motor coordination. The ex vivo BBB assay was performed according to ethics approval KEK-ZH-Nr: 2015-0589 for Zurich University of Applied Sciences, Waedenswil, Switzerland. Human GBM explant bioreactor experiments were performed according to the ethics approval Req-2019-00553 at the University Hospital Basel, Basel, Switzerland.

All patients provided written informed consent for material and data collection, analysis, and sharing of the results.

## Recombinant proteins

*Human and murine IL-12Fc variants* were expressed in-house in adherent HEK239T cells after calcium phosphate transient transfection according to published protocols[22], or were custom-produced in suspension WuXianXpress HEK293 cell cultures (WuXi Biologics). FcRn low-affinity variants were generated by exchanging isoleucine 253 and histidines 310 and 435 with amino acids bearing inert side chains, or with similar side chain length and flexibility. Certain combinations of amino acid substitutions resulted in proteins with a tendency to precipitate and to lose protein G affinity (Table S1). This was the case if both mutations I253N and H310A were present. Cell culture supernatants were cleared before loading on the affinity chromatography resins: murine IL-12Fc on Protein A sepharose (Cytiva, cat. 29048576) and all other proteins on Protein G sepharose (Amsbio, cat. 6511-5), then concentrated and buffer-exchanged on spin columns with a 30 kDa molecular weight cut-off (Sartorius, cat. VS0221). The recombinant proteins were then further purified by ion-exchange chromatography using diethylaminoethylcellulose columns (Cytiva, cat. 17505501) and size exclusion chromatography columns (Superose 6 Increase 10/300GL, Cytiva, cat. 29091596) and eluted into histidine buffer (20 mM histidine: Sigma-Aldrich, cat. 84088; 150 mM NaCl: Sigma-Aldrich, cat. S5886; 0.05% Tween 20: Sigma-Aldrich, cat. P1379; pH = 6.0). Homodimer fractions were pooled and concentrated with ultrafiltration spin columns (Sartorius, cat. VS0221). Protein purity was validated by SDS-PAGE with Coomassie blue staining (Lubio Science, cat. LU0010000).

A biosimilar of *efgartigimod* (MST-HN Fc with "Abdeg" mutations M252Y/S254T/T256E/H433K/N434F[28,29] and starting at residue D221 in the hinge region, EU numbering), was produced using the pcDNA3.4 expression plasmid (Thermo Fisher Scientific, cat. A14697) transfected (Expifectamine, Transfection Kit; Thermo Fisher Scientific, cat. A14524) into HEK293 suspension cultures (Expi293F, Thermo Fisher Scientific, cat. A14527), followed by protein A sepharose affinity chromatography. Aggregates were removed using size exclusion chromatography as above.

*Unmodified recombinant mouse IL-12* was purchased from Miltenyi Biotec (cat. 130-096-795). *Unmodified recombinant human IL-12* was purchased at Peprotech (cat. 200-12) or expressed in HEK293 cells and purified through a C-terminally fused FLAG tag on the p35 subunit.

## Animals

C57BL/6J mice were obtained from Charles River (Sulzfeld, Germany) or Janvier Labs (Le Genest-Saint-Isle, France). B6.Cg-*Fcgrt*^tmlDcr^ Tg (FCGRT)32Dcr/DcrJ (hFcRn Tg32) mice were purchased from The Jackson Laboratory (stock number 014565) and bred in-house. C3H/HeJ (Heston) mice were purchased from The Jackson Laboratory (stock number 000659). Euthanasia was performed by controlled $CO_2$ asphyxiation.

## Cell lines

GL-261 cells were provided by A. Fontana (Institute of Experimental Immunology, University of Zurich, Zurich, Switzerland) and cultured in DMEM (Sigma-Aldrich, cat. D6429) supplemented with 10% heat-inactivated fetal bovine serum (FBS; Pansera, cat. 2602-P260328) and 1% L-glutamine (Thermo Fisher Scientific, cat. 25030081). GL-261 cells were stably transfected with pGL3-ctrl (Promega, cat. E174A) and pGK-Puro (Addgene plasmid #11349[69]) and selected with puromycin (Sigma-Aldrich, cat. P7255) to generate luciferase-stable GL-261:luc cells. SB28 cells were provided by H. Okada, University of California San Francisco. Cells were cultured at 37 °C 5% $CO_2$ in DMEM (Sigma-Aldrich, cat. D6429) supplemented with 10% heat-inactivated FBS (Pansera, cat. 2602-P260328), 1% L-glutamine (Thermo Fisher

Scientific, cat. 25030081) and 1% Penicillin-Streptomycin (Thermo Fischer Scientific, cat. 15140122). All cell lines were regularly tested for mycoplasma contamination.

## Tumor cell injection

For glioma inoculation, 6-to-10-week-old mice were anesthetized using Fentanyl 0.05 mg/kg BW (Helvepharm AG, cat. 7818566), Midazolam 5 mg/kg BW (Roche Pharma AG, Dormicum), and Medetomidin 0.5 mg/kg BW (Orion Pharma AG, Dormitor), injected via the intraperitoneal (i.p.) route. A total of 20,000 GL-261:luc cells[22] or 1600 SB28 cells[42] in 2 μL of PBS were injected intracranially into the right striatum using a stereotactic robot (Neurostar, Stoelting 51615) and a blunt-ended needle with syringe (Hamilton) through a burr hole made using a 0.9 mm drill bit (Hager & Meisinger, cat. 1RF HP 009). The stereotactic coordinates were 1.5 mm lateral and 1 mm frontal of the bregma. The needle was lowered into the burr hole to a depth of 4 mm below the skull surface and then retracted by 1 mm, before starting the injection at a rate of 1 μL/min. After leaving the needle in place for 2 min, it was retracted at 1 mm/min. The burr hole was closed with bone wax (Braun, cat. 1029754) and the skin was sealed with tissue glue (Zoetis, cat. 32046). Anesthesia was reversed using a mixture of Flumazenil 0.5 mg/kg BW (Labatec Pharma AG) and Buprenorphine 0.1 mg/kg BW (Indivior Schweiz AG), and Atipamezol 2.5 mg/kg BW (Janssen), administered subcutaneously. Buprenorphine 0.1 mg/kg BW (Indivior Schweiz AG), administered subcutaneously (during day) or 9.1 mg/L of drinking water (during night), was used for analgesia.

## Protein administration by osmotic minipumps

Administration using osmotic mini pumps was performed as described previously[22]. Briefly, osmotic pumps (0.25 μL/h; Alzet, model 2004) were filled with murine IL-12Fc (12.5 μg/kg/24 h) or PBS alone and primed at 37 °C in PBS. Mice were handled as described for tumor injection. The infusion cannula was lowered through the burr hole made for tumor cell injection to a depth of 3 mm.

## Protein administration by intraparenchymal injection

Mice were handled as described for tumor cell injection above. CED into the brain was performed as previously published[17]. Briefly, mice underwent injection using a Hamilton syringe 10 μL (Hamilton, cat. 7635-01) with catheters made using a 27 G blunt-ended needle (Hamilton, cat. 7768-01) with a 1 mm step at the tip made of fused silica with an internal diameter of 0.1 mm and wall thickness of 0.0325 mm (Postnova, cat. Z-FSS-100165). For non-tumor-bearing mice, a burr hole was drilled at a position 1 mm frontal and 2 mm lateral of bregma. The catheter was lowered into the burr hole to a depth of 3.5 mm below the dura surface. Injection was performed in a volume of 1 μL at 0.2 μL/min, then 2 μL at 0.5 μL/min, and 2 μL at 0.8 μL/min. After leaving the needle in place for 2 min, it was retracted at 1 mm/min. Injection mixes were prepared in PBS or histidine buffer (20 mM histidine: Sigma-Aldrich, cat. 84088; 150 mM NaCl: Sigma-Aldrich, cat. S5886; 0.05% Tween 20: Sigma-Aldrich, cat. P1379; pH = 6.0).

## Systemic protein administration

For in vivo blockade of VLA-4, GL-261 tumor-bearing mice underwent i.p. injection with six doses of 100 μg of anti-VLA-4 (clone PS/2, BioX-Cell, cat. BE0071) or the corresponding isotype control (clone LTF-2, BioXCell, cat. BE0090), at days 21, 25, 28, 32, 35, and 39 post-tumor inoculation. Similarly, non-tumor-bearing mice underwent i.p. injection with three doses of 100 μg of anti-VLA-4 or isotype control, with the initial dose 1 day before CED, followed by subsequent doses at day 2 and day 5 post-CED.

To study the systemic influence of FcRn inhibitor, MST-HN Fc was intravenously (i.v.) injected into hFcRn Tg32 mice at 100 μg per mouse. Twenty-four hours post-injection, CED was performed with

either hIL-12Fc WT or NHQ into the right striatum. Six hours post-CED, the mice were euthanized, and plasma and brain tissue were collected for further analysis.

## In vivo bioluminescence imaging

Tumor-bearing mice underwent i.p. injection with D-Luciferin (Potassium salt, 150 mg/kg BW, Swissslumix, cat. SL30101) and 15 min later were anaesthetized with isoflurane. Luminescence was recorded using the Xenogen IVIS Lumina III (PerkinElmer) imaging system and data analyzed using Living Image 4.7.1 software (PerkinElmer). A circular region of interest (ROI; 1.5 cm diameter) was defined around the tumor site, and the photon flux of this region was read out and plotted (p/s/cm²/sr).

## Chemotherapy

Mice bearing GL-261 brain tumors were treated by injection with temozolomide (Adipogen, cat. CDX-T0072-M250) at 50 mg/kg, administered i.p. Chemotherapy was diluted in PBS 5% DMSO (Sigma-Aldrich, cat. D2650) immediately before injection. Treatment was started on day 23 and continued daily to day 27 (five doses in total).

## Radiotherapy

Mice bearing SB28 brain tumors were anesthetized as described above for tumor cell injection and positioned within lead shields with adjustable 5 mm pinholes to expose only the tumor-injected side of the brain. Ten Gy of X-ray radiation was delivered to the tumor based on dosimetry evaluation at a rate 1.28 Gy/min using an RS2000 irradiator (Rad Source Technologies).

## Brain protein isolation

After euthanasia and removal of the skullcap, the cerebellum and olfactory bulbs were discarded, the brain hemispheres were separated along the midline, and the injected (ipsilateral) hemisphere was frozen at −20 °C. Brain lysates were prepared by homogenization in ice-cold lysis buffer (Cell Signaling, cat. 9803S) to a final concentration of 2x lysis buffer with protease inhibitors (Roche, cat. 11836153001). 0.1 mL of lysis buffer was added per 10 mg of brain tissue. Brain tissue was minced using scissors, then passed through a 20 G needle and finally sonicated for 20 s. Samples were centrifuged at 10,000 × g at 4 °C for 10 min, and supernatants were stored at −80 °C. Protein concentration was measured using a Pierce BCA assay kit (Thermo Fisher Scientific, cat. 23225) and these data were used to correct for the protein extraction efficiency within each experiment (corrected ELISA concentration = (ELISA concentration) × (average total protein concentration in all the samples in the experiment)/(total protein concentration of the sample)).

## Isolation of plasma and serum

For plasma isolation, blood was collected from the tail vein or, upon sacrifice, by heart puncture into dried $K_2EDTA$ tubes (BD, cat. 365975) and centrifuged for 5 min at 10,000 × g. Plasma was transferred to 96-well microplates (Thermo Fischer Scientific, cat. AB-0600) and frozen at −20 °C. For serum isolation, blood was collected into a micro-centrifuge tube and centrifuged for 5 min at 10,000 × g. Serum was transferred to 96-well microplates (Thermo Fischer Scientific, cat. AB-0600) and frozen at −20 °C.

## IL-12 detection in the brain and blood after injection into the brain

Samples were diluted in blocking buffer (PBS containing 0.05% Tween 20: Sigma-Aldrich, cat. P1379; and 0.1% bovine serum albumin (BSA): Sigma-Aldrich, cat. A9647) and IL-12 levels were measured by ELISA for human IL-12p70 (Mabtech: Fig. 2C and S2G, cat. 3455-1H-6; or Thermo Fischer Scientific: Figs. 1, 2D–F, S1, S2H–J, cat. 88-7126-88).

## Bead-based cytokine array

Plasma levels of murine IL-12, IFNγ, and CCL-2 were measured using the Legendplex Mouse Inflammation Panel (Biolegend, cat. 740446) following the manufacturer's protocol, with data acquired on an LSRII Fortessa (BD). IFNγ levels over 28 ng/mL shown in Fig. 4 were extrapolated by linear regression; all other values were interpolated from the corresponding standard curves.

## Plasma ALT measurement

Levels of ALT were measured in plasma samples using the AU480 Chemistry Analyzer (Beckman Coulter) at the Zurich Integrative Rodent Physiology facility.

## Flow cytometry

The following antibodies were used for flow cytometry analysis of immune infiltrate: anti-CD45 (clone 30-F11), anti-CD3 (clone 17A2 or 145-2C11), anti-CD4 (clone GK1.5), anti-CD8α (clone 53-6.7), anti-CD11b (clone M1/70), anti-FoxP3 (clone 150D or MF23), anti-PD-1 (clone RPM1-30 or 29F.1A12), anti-IFNγ (clone XMG1.2), anti-I-Ab (clone AF6-120.1 or M5/144.15.2), anti-CD19 (clone 6D5), anti-Ly6G (clone 1A8), anti-Ki67 (clone 16A8), anti-NK1.1 (clone PK136), anti-CD69 (clone H1.2F3) and anti-TNFα (clone MP6-XT22). Cells were also stained with the live/dead discriminator Zombie Aqua (BioLegend, cat. 423101) or NIR (BioLegend, cat. 423106). Mice were euthanized and perfused with ice-cold PBS. Tumor-bearing hemispheres were minced in digestion medium containing RPMI (Sigma-Aldrich, cat. R1383) with 0.2 mg/mL Collagenase D (Roche, cat. 7002219), 0.5 mg/mL DNase I (Roche, cat. 10104159001) and 10% FBS (Pansera, cat. 2602-P260328) and incubated for 30 min at 37 °C. Tissue homogenate was filtered through a 70 μm pore diameter cell strainer (Falcon, cat. 10788201), centrifuged, and separated on a Percoll gradient (Cytiva, cat. 17-0891-01). Cells were either labeled directly or first restimulated before labeling for intracellular cytokines for 4 h at 37 °C in RPMI (Sigma-Aldrich, cat. R1383) containing 10% FBS (Pansera, cat. 2602-P260328), 1 μL/mL Brefeldin A (BD, cat. 555029), 500 ng/mL Ionomycin (Thermo Fischer Scientific, cat. I24222) and 50 ng/mL phorbol-12-myristate-13-acetate (Sigma-Aldrich, cat. P1585). Antibody labeling was performed in the presence of anti-mouse CD16/32 (TruStain FcX, Biolegend, cat. 101320). A full list of antibodies is provided as Supplementary Data 1. For absolute cell counting of blood leukocytes, 5 μL of counting beads (Thermo Fischer Scientific, cat. C36950) per sample was added prior to data acquisition. For intracellular labeling, surface labeling was followed by a 30 min incubation step at 4 °C in Fix/Perm buffer from the FoxP3/Transcription factor staining buffer kit (Thermo Fischer Scientific, cat. 00-5523-00) following the manufacturer's instructions. Briefly, samples were washed with PermWash buffer prior to adding antibodies for intracellular staining. Samples were washed again in permwash buffer, resuspended in PBS, filtered through a 35 μm pore diameter strainer, and data acquired using an LSR II Fortessa (BD) or 5L Cytek Aurora (Cytek Biosciences) flow cytometer.

For FcRn expression analysis of iPSC-derived BMECs by flow cytometry, cells were trypsinized with TrypLE Select (ThermoFisher Scientific, cat. 12604021), washed, and fixed with 3% formaldehyde (Sigma-Adrich, F8775) prior to being permeabilized with 0.1% Saponin (Sigma-Aldrich, cat. 558255). Next, cells were incubated with AF488-conjugated anti-FcRn antibody (clone 937508; R&D Systems, cat. IC8639, 1:100) in PBS-Saponin buffer for 30 min at room temperature. Data were acquired on a FACS Aria III cytometer (BD).

## Histology

Euthanized mice were transcardially perfused with ice-cold PBS followed by 4% formaldehyde (Sigma-Adrich, cat. F8775) in PBS at a rate of 4 mL/min. The brains were removed, incubated for 16 h in 4% formaldehyde (Sigma-Adrich, cat. F8775) in PBS, dehydrated, and embedded in paraffin. Upon cutting onto glass slides, sections of mouse brains were labeled using a Ventana Discovery Ultra automated

slide preparation system, with antibodies against CD4 (clone 4SM95; Thermo Fischer Scientific, cat. 14-9766-82, 1:50) and CD8 (clone 4SM15; Thermo Fischer Scientific, cat. 14-0808-82, 1:50).

Immunofluorescence staining for FcRn-silenced IL-12Fc was performed as follows. Mice were euthanized and transcardially perfused with PBS, followed by 4% formaldehyde (Sigma-Adrich, cat. F8775) in PBS. Brains were dissected and additionally fixed with 4% formaldehyde in PBS at 4 °C for 24 h. Subsequently, brains were washed with 15% sucrose for 60 min and transferred to 30% sucrose at 4 °C. After 24 h, brains were frozen on dry ice. Free-floating sections (25 μm) were stained using rat anti-mouse IL-12p40 (clone C17.8, BioXCell, cat. BE0051, 1:200) and polyclonal goat anti-rat IgG (H + L) antibody coupled with Alexa Fluor 647 (Jackson Immuno Research, cat. 112-605-167, 1:100) and counterstained with DAPI (Sigma-Aldrich, 269298). SB28 tumor cells were further stained with polyclonal rabbit anti-GFP antibody (Abcam, cat. Ab290, 1:200) and polyclonal donkey anti-rabbit IgG (H + L) antibody coupled with FITC (Jackson Immuno Research, cat. 711-095-152, 1:100). Slides were scanned using an Akoya Vectra Polaris scanner (Akoya Biosciences) and data were analyzed using QuPath[70] and Fiji[71].

For histopathological analysis of organs from the toxicology experiment in C3H/HeJ mice, animals were sacrificed, blood was collected from the right heart ventricle, spleens were removed, and mice were immersed in 4% formaldehyde (Sigma-Adrich, cat. F8775) in PBS for 48 h, after which they were transferred to 70% ethanol. Afterwards, spleens, brains, livers, lungs, kidneys, hearts, and large intestines were dissected and processed for formalin-fixation and paraffin-embedding. Sections of 2–3 μm thickness were mounted on glass slides and routinely stained with hematoxylin and eosin. Furthermore, consecutive sections of brain samples (2–3 μm) were used for immunohistochemistry (IH). IH was performed using the horseradish peroxidase (HRP) method to identify macrophages/activated microglial cells (Iba-1), to highlight astrocytes (glial fibrillary acidic protein, GFAP), apoptotic cells (cleaved caspase-3), and neurons (NeuN). Briefly, after deparaffination, sections underwent antigen retrieval in citrate buffer (pH 6.0; for Iba-1 and GFAP labeling) or Tris/EDTA buffer (pH 9; for cleaved caspase-3 and NeuN) for 20 min at 98 °C, followed by incubation with the primary antibodies diluted in dilution buffer (Agilent, cat. S080983-2). Used antibodies were polyclonal rabbit anti-Iba-1 (Wako, cat. 019-19741, 1:750), polyclonal rabbit anti-GFAP (Agilent, cat. Z033429-2, 1:600), polyclonal rabbit anti-mouse cleaved caspase-3 (Sigma-Aldrich, cat. RM250, 1:200) or mouse anti-NeuN (clone A60; Sigma-Aldrich, cat. ZMS377, 1:4000). This was followed by the blocking of endogenous peroxidase (peroxidase block; Agilent, cat. S202386-2) for 10 min at RT and incubation with the appropriate secondary antibodies/detection systems: EnVision+ HRP (Agilent, cat. K400311-2) for Iba-1 and cleaved caspase-3 or MACH4 HRP (Biocare Medical, cat. M4U534) for GFAP and NeuN. The whole staining procedure was performed using an autostainer (Agilent or Ventana). Sections were subsequently counterstained with hematoxylin. Sections incubated without the primary antibodies served as negative controls. All histological sections were semi-quantitatively evaluated by a board-certified veterinary pathologist in a blinded manner.

## FcRn expression analysis

For transcription of *Fcgrt* and *Pecam1* in GL-261 cells, RNA was isolated from tumor-bearing murine brain hemispheres using the QIAshredder (Qiagen, cat. 79656) and Qiagen RNeasy mini-RNA extraction kit (Qiagen, cat. 74104) according to the manufacturer's instructions. For cDNA synthesis, 1 μg of RNA template was mixed with 50 μM Oligo(dT)12-18 primers (Thermo Fischer Scientific, cat. 18418012), 10 mM dNTP mix (Promega, cat. U1511), and H$_2$O in a total volume of 15 μL, then heated for 5 min at 65 °C. After keeping on ice for 1 min, mix was added to each primer-RNA mix and incubated at 52 °C for 10 min and 80 °C for 10 min before storing the cDNA at −20 °C. SYBR Green (Bio-Rad, cat. 1725270)

was used according to the manufacturer's instructions, with 450 nM of primers and 10 ng of cDNA template input in a volume of 5 μL per reaction. Data were analyzed using the Bio-Rad CFX Software. Primers (5'-3'): mFcRn_fw: TGTCTGTCGTCTTGGACTGG; mFcRn_rev: AGGCAGGAGGACCAACAAG; mCD31_fw: GAGCCAGCAGTATGAGGACC, mCD31_rev: GGCGATGACCACTCCAATGA HPRT_fw: GACCGGTCCCGTC ATGC, HPRT_rev: TCATAACCTGGTTCATCATCGC. Data on FcRn and PECAM1 (CD31) protein abundance were extracted from published datasets S1 A-H[20].

## Ex vivo BBB assay

iPSC maintenance and differentiation to BMECs: The iPSC cell line was generated from human dermal skin fibroblasts from a healthy donor, by lentiviral transduction of Oct4, Sox2, Klf4, and cMyc according to established protocols[72]. The iPSC cell line was cultured in the Essential 8 media system (Thermo Fisher Scientific, cat. A1517001) on vitronectin-coated plates (5 μg/cm² vitronectin, Thermo Fisher Scientific, cat. A31804) at 37 °C and 5% CO$_2$ in a humid atmosphere. Medium was supplemented with Thiazovivin (0.4 μM; Miltenyi Biotec, cat. 130-14-461). The differentiation of iPSCs to BMECs was performed according to ref. 44 with minor variations: Y-27632 was replaced by 0.4 μM Thiazovivin, and a 1:2 split of the cells into Cultrex-coated plates (R&D Systems, cat. 3434-050-RTU) was carried out on day 3 of the differentiation. For the second phase of the differentiation (days 4–9), the cells were grown in human endothelial serum-free medium (hESFM; Thermo Fisher Scientific, cat. 11111044) to achieve full differentiation of iPSCs to BMECs. Transwell assay: Thereafter, the cells were trypsinized, washed with PBS or hSFM, and seeded onto transwell filters (12-well format, 1.1 cm² polyethylene terephthalate membranes with 0.4 μM pores; Sarstedt, cat. 83.3931.041). The inserts were previously coated with a mixture of collagen IV (4 μg/cm²; Sigma-Aldrich, cat. CC076) and fibronectin (4 μg/cm²; Sigma-Aldrich, cat. F0895) for 1 h at 37 °C. 500 μL of medium was applied to the apical chamber and 1.5 mL of medium to the basolateral chamber.

Immediately prior to the transwell assay, trans-endothelial electrical resistance (TEER) was measured at the peak value using an EVOM voltmeter with STX2 electrodes (World Precision Instruments). Raw resistance values were adjusted by subtracting the resistance measured across an empty filter, and final TEER values were calculated as the measured resistance value (Ω) multiplied by the surface area (1.1 cm²). Samples of transwells with a TEER value below 1500 Ω × cm² were not used for the transport assays.

For the permeability studies, IL-12, IL-12Fc WT, or IL-12Fc NHQ at a final concentration of 100 ng/mL were either applied to the apical or basolateral chamber. The first sample from both compartments was taken after the application, and the second after 24 h. The amount of IL-12 or IL-12 variants was determined by anti-IL-12p70 ELISA (ThermoFisher Scientific, cat. 88-7126-88). The apparent permeability coefficient (Papp) was then calculated using the following formula for the transport from the basolateral compartment to the apical compartment: Papp(cm/s) = Va/(ACb0)x(dCa/dt) where Va is the volume in the apical chamber, A is the surface area of the filter membrane (1.1 cm²), Cb0 is the initial concentration in the basolateral compartment and dCa/dt is the change of concentration over time in the apical chamber. The Papp for the transport from the apical side to the basolateral side was calculated accordingly.

## Explant bioreactors

Biopsy samples from contrast medium–enhancing 5-ALA–positive tumor centers were collected during surgery via intraoperative navigation. Tissue acquisition was documented by intraoperative imaging (taking into consideration 5-ALA positivity and neuronavigation). In total, nine tumor samples were included in this study in two series (see Table S3). Patient diagnosis, pretreatments, including steroid exposure, IDH and MGMT methylation status characteristics are

summarized in Table S3. Briefly, intact GBM tissues were placed into ice-cold PBS immediately following tumor resection and were directly transferred to the laboratory. Fragments of tumor tissue, devoid of blood or necrosis, were cut into 50–100 mm$^3$ fragments. Tissue fragments were placed between two silicone adapters and ethylene-tetrafluoroethylene copolymer mesh grids, which were applied to the top and bottom of the adapters and were placed into U-CUP perfusion chambers (Cellec Biotek AG). The perfusion medium consisted of a 1:1 mix of Neurobasal-A medium (Thermo Fisher Scientific, cat. 10888022) and Dulbecco's modified Eagle's medium/F12 medium (Sigma-Aldrich, cat. 11320033) supplemented with nonessential amino acids (Sigma-Aldrich, cat. M7145), 1 mM sodium pyruvate (Sigma-Aldrich, cat. S8636), 44 mM sodium bicarbonate (Thermo Fisher Scientific, cat. 25080094), 25 mM HEPES (Thermo Fisher Scientific, cat. 15630056), 4 mM l-alanyl-l-glutamine (Thermo Fischer Scientific, cat. 35050038), antibiotic-antimycotic (Thermo Fisher Scientific, cat. 152400662), human recombinant epidermal growth factor (20 ng/mL; Thermo Fisher Scientific, cat. PHG0314), human recombinant fibroblast growth factor (20 ng/mL; Peprotech, cat. 10018B), heparin sulfate (10 ng/mL; Stemcell Technologies, cat. 07980), and 5% human serum (Sigma-Aldrich, cat, H4522). Human recombinant epidermal growth factor, human recombinant fibroblast growth factor, and 5% human serum were added immediately before perfusing the bioreactor systems. The bioreactor set up was either perfused with untreated medium or supplemented with 100 ng/mL of IL-12Fc NHQ, with the perfusion flow rate set at 0.47 mL/min to achieve a superficial flow velocity of 100 μm/s at 37 °C, 95% humidity, and 5% $CO_2$. Samples were collected on day 7 (series 1) or days 4 and 7 post-treatment (series 2). For sampling, 500 μL of supernatants were withdrawn from each bioreactor using a cannula, centrifuged at $10,000 \times g$ for 5 min at 4 °C, and supernatants were collected and frozen at −80 °C. All supernatants were used for proteomics analysis.

## Proteomics analysis
Supernatants from patient explant bioreactors were analyzed using the Target 96 Immuno-Oncology panel (Olink) according to the manufacturer's instructions at the UZH Olink core facility. For data analysis, the following processing steps were applied to the pre-normalized and log$_2$-scaled protein expressions: analyte reads containing more than 75% of samples below the limit of detection were removed in line with recommendations by Olink; next, all samples with technical replicates were collapsed by transforming the NPX-values to the linear scale, then averaging them, before transforming back to the log$_2$-scale. Six missing data points were reported for IL-4, but all other panel, sample, and assay metrics were within the acceptable norms.

For exploratory analysis, the data were scaled and centered to z-scores. The assay IL-4 was excluded for singular value decompositions of the replicates' expressions. The principal components of the transformed counts were calculated using the prcomp function of the stats package version 4.3.1, written in the statistical computing software R.

Differential expression analysis was performed with the linear regression tools from the R package limma version 3.56.2[73]. The differential expression statistics were estimated with two-sided, moderated t-tests. The linear model included the additive factors Patient, Treatment, and Time. The function limma::arrayWeights was used for evaluating quality weights of the individual samples[74]. The regression details include a robust hyperparameter estimation and a calculation of the protein mean–variance trend[75].

The gene set analysis focuses on the hallmark gene sets[76] defined for the Molecular Signatures Database[77] of the Broad Institute. These gene sets were retrieved with the msigdbr package version 7.5.1[78] on 2023-11-28.

Statistical relevance was determined by rotational gene set tests[79] implemented in the limma::mroast function. The rotational tests incorporated the same design formula as in the linear model of the protein expression. The quality weights were carried over as well.

False discovery rate (FDR) adjustment was accomplished using the Benjamini–Hochberg method[80].

The version of the R computing language is 4.3.1 (2023-06-16)[81].

## SDS-PAGE
One microgram of protein was diluted in 12 μL of PBS and 3 μL of Laemmli loading dye (Bio-Rad, cat. 1610747) with or without 10% 2-Mercaptoethanol (Bio-Rad, cat. 1610710). The samples were heated at 95 °C for 5 min before loading into a PAGE 4-12% gradient Tris-MOPS gel (GenScript, cat. M00656). Dual Color Protein Standard (Bio-Rad, cat. 1610374) was used. Gels were stained with Coomassie blue staining (Lubio Science, cat. LU0010000).

## FcRn binding measured by ELISA
IL-12Fc variants or a recombinant human IgG4 anti-GFP antibody (clone 515; AbD Serotec, cat. HCA195), which served as a control, were used to coat ELISA plates at 2 μg/mL, before application of the assay buffer (10 mM Citric acid: Sigma-Aldrich, cat 251275; 150 nM NaCl: Sigma-Aldrich, cat. 84088; 0.05% Tween 20: Sigma-Aldrich, cat. P1379, pH = 6.0) with 0.1% BSA (Sigma-Aldrich, cat. A9647) for blocking for 1 h at RT. Biotinylated FcRn (Immunitrack, cat. 1TF01) was incubated for 2 h at RT at increasing concentrations in assay buffer. FcRn was detected using streptavidin-coupled horseradish peroxidase (Mabtech, cat. 3310-9-1000) and colorimetric substrate (Thermo Fischer Scientific, cat. SB01). Absorbance was measured at 450 nm.

## FcRn binding measured by surface plasmon resonance (SPR)
SPR was performed using the ProteOn XPR36 System (Bio-Rad), coating human recombinant biotinylated FcRn (Immunitrack, cat. 1TF01) on a ProteOn NLC sensor chip (Bio-Rad, cat. 176-5021). Tested coating density was in the range 0-600 RU, with optimal signal-to-noise ratio at 100-115 RU signal for antibodies and 250-260 RU for IL-12Fc constructs. Analytes were injected in the range 0–6075 nM. Dissociation constants ($K_D$) were calculated based on the binding equilibrium analysis. Running buffer was histidine buffer (20 mM histidine: Sigma-Aldrich, cat. 84088; 150 mM NaCl: Sigma-Aldrich, cat. S5886; 0.05% Tween 20: Sigma-Aldrich, cat. P1379; pH = 6.0) and recovery buffer was 20 mM Tris-HCl (Sigma-Adlrich, cat. 10812846001) pH = 7.5 with 150 mM NaCl (Sigma-Aldrich, cat. S5886) 0.05% Tween 20 (Sigma-Aldrich, cat. P1379).

## Thermal stability assay
The thermal stability assay was performed using SYPRO Orange (Thermo Fisher Scientific, cat. S6650). Four microgram of protein at a concentration of 0.4 μg/μL per was mixed with SYPRO Orange solution diluted in PBS or artificial CSF (aCSF; 125 mM NaCl: Sigma-Aldrich, cat. S5886; 26 mM NaHCO$_3$: Sigma-Aldrich, S6014; 1.25 mM NaH$_2$PO$_3$, Sigma-Aldrich, cat. 04273; and 2.5 mM KCl, Sigma-Aldrich, cat. P5405) and measured on a thermocycler (Bio-Rad) set to 2 min at 20 °C, followed by a 0.2 °C increase/30 s until 95 °C. Fatty acid-free albumin was used as a control.

## HEK-Blue IL-12 bioactivity assay
HEK-Blue IL-12 cells were plated according to the manufacturer's protocol (InvivoGen). Cells were incubated with increasing amounts of IL-12, IL-12Fc WT, or variants designed for reduced FcRn affinity for 17 h. Culture medium was collected and incubated for 2 h in the presence of Quanti-Blue detection reagent (InvivoGen, cat. rep-qbs). Absorbance was measured at 640 nm.

## IFNγ production by murine splenocytes stimulated with mIL-12hFc
Spleens were mashed through a 70 μm pore diameter mesh (Falcon, cat. 10788201), then cells were washed, and erythrocytes were

removed using a lysis buffer (BioLegend, cat. 420301). Cells were seeded into a 96-well cell culture plate (Thermo Fischer Scientific, cat. AB-0600) at 100,000 cells per well and stimulated with anti-CD3ε (Thermo Fischer Scientific, cat. 16-0032-82) and anti-CD28 (Thermo Fischer Scientific, cat. 16-0281-82), at a concentration of 1 μg/mL each. IL-12 was added at varying concentrations and incubated for 2 days at 37 °C, 95% humidity, and 5% $CO_2$. Readout was by mouse IFNγ ELISA according to the manufacturer's protocol (BD, cat. 555138).

### IFNγ production by lymphocytes stimulated with IL-12Fc
Human peripheral blood mononuclear cells (PBMCs) were isolated from buffy coats (SRK Blutspende Zürich) using ficoll-paque (Cytiva, cat. 17-5422-02) density gradient centrifugation. Cells were frozen in 50% RPMI (Sigma-Aldrich, cat. R1383), 40% FBS (Pansera, cat. 2602-P260328), 10% DMSO (Sigma-Aldrich, cat. D2650). Thawed PBMCs were plated in a U-bottom 96-well plate (Thermo Fischer Scientific, cat. AB-0600) at $10^6$ cells per well and incubated for 3-4 h before the addition of increasing concentrations of IL-12 variants in the presence of 100 ng/mL anti-CD3 antibody (BioXCell, cat. BE0001-2). After 24 h, IFNγ levels in supernatant were measured by ELISA following the manufacturer's instructions (Mabtech, cat. 3420-1H-20).

### Statistics and reproducibility
For all non-survival analyses of two experimental groups, a two-tailed Student's *t*-test was performed. For non-survival analyses of three or more experimental groups with an all vs all statistical testing, a one-way ANOVA with Tukey's correction for multiple testing was performed. For non-survival analyses of three or more experimental groups with testing to control group, a one-way ANOVA with Dunnett's correction for multiple testing was performed. For non-survival analyses of longitudinal data, two-way ANOVA with Sidak's multiple comparison test was used. Only statistically significant differences (*p* values < 0.05) were shown in all the graphs but Fig. 2D. For statistical analysis of Kaplan-Meier survival curves, a Log-rank (Mantel–Cox) test was used to calculate the *p* values indicated in respective experiments. Group sizes were determined based on power analysis, and experiments were pooled as indicated to reach the required number of animals per group. All animal experimenters were blinded with regard to the administration of treatment, scoring, and sample ID in subsequent analyses, until statistical assessment was concluded. Outliers in animal data were identified and removed based on Grubbs' test, as noted in the figure legends. No outlier removal was performed in human datasets. In mouse tumor experiments, group allocation has been performed as follows: on day 20 after implantation of GL-261:luc, or day 4 after implantation of SB28 glioma cells, animals were distributed into experimental groups based on tumor size as measured by in vivo bioluminescence imaging (BLI). Inclusion criteria in the GL-261 model were a tumor ROI signal $10^4$ to $10^8$ p/s/cm²/sr. Animals from both sexes have been used according to availability and in adherence to 3R principles. Sex was not considered in the study design, and no sex-related analyses were performed due to a lack of statistical power. All quantitative analysis was performed with MS Excel (Microsoft) or GraphPad Prism version 10.2.3 for Mac OS X (GraphPad Software, Inc.). Flow cytometry data were analyzed using FlowJo Software v10 (BD).

### Reporting summary
Further information on research design is available in the Nature Portfolio Reporting Summary linked to this article.

### Data availability
The flow cytometry data generated in this study have been deposited in the Zenodo database under accession code 15263559 [https://doi.org/10.5281/zenodo.15263559, https://zenodo.org/records/15263560]. The human proteomics data generated in this study have been deposited in the Zenodo database under accession code 15282130 [https://doi.org/10.5281/zenodo.15282129, https://zenodo.org/records/15282130]. All other data generated in this study are provided in the supplementary information file and source data file. Source data are provided with this paper.

### Code availability
Proteomic analyses were performed using freely available R packages. Plots and graphs were generated with software and pipelines as cited in the respective methods section.

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

## Acknowledgements

The authors would like to thank Dr. Anja Heider (SIAF, Davos, Switzerland) for excellent assistance during secretome analysis experiments, Dr. Jürgen Besserer (Hospital Hirslanden, Zurich, Switzerland) for X-ray calibration and dosimetry, Tobias Weiss and Michael Weller (University Hospital Zurich, Zurich, Switzerland) for sharing the temozolomide treatment SOPs and Mark Gradwell, Irma Wagenaar and Pavel Savitskiy (University of Southampton, Southampton, UK) for generating recombinant MST-HN Fc. We wish to thank Dr. Lucy Robinson of Insight Editing London for critical review and editing of the manuscript draft prior to submission. This work was supported through grants of the Novartis Foundation for Medical-Biological Research (16C231), Swiss Life Jubiläumsstiftung (1283-2021) and Swiss Cancer Research (KFS-3852-02-2016, KFS-4146-02-2017, KFS-5306-02-2021) and Lindonlight Collective LLC (GR-24-010) to J.v.B. Further funding was provided by Swiss National Science Foundation (SNSF; NRP79, 407940_206465) to J.v.B., by the Entrepreneur Program of University of Zurich (BIOEF19-014) to M.B. and by a SNSF and Commission for Technology and Innovation (CTI) BRIDGE Proof of Concept grant (project number 40B1-0_177300) to L.S. Moreover, part of the work was supported by a SNSF Professorial Fellowship (PP00P3_176974 to G.H. and PP00P3_144823 to M.A.S.); Innosuisse (56545.1 IP-LS) to J.R.; the ProPatient Forschungsstiftung, University Hospital Basel (Annemarie Karrasch Award 2019); the Department of Surgery, University Hospital Basel, and by The Brain Tumour Charity Foundation, London, UK (GN-000562) to G.H.

## Author contributions

J.v.B. supervised the study. J.v.B. designed, analyzed, and interpreted data, and N.T., B.T., U.S., L.S., and M.B. designed, performed, and analyzed experiments. S.D., T.S., M.F.R., L.v.B., and M.McD. performed and analyzed experiments. F.S., S.S., and C.P.Z. analyzed experiments and interpreted data. J.R., T.B., G.H., I.Z., P.E., and M.A.S. provided infrastructure and data, helped with data interpretation, and provided conceptual input. H.O. and E.S.W. provided reagents, helped with data interpretation, and provided conceptual input. M.B., L.S., B.T., and J.v.B. wrote the original manuscript. All the authors read and reviewed the manuscript.

## Competing interests

The here described compartment-locked IL-12Fc is covered by patent applications WO2020201167A1, WO2020201168A1, and EP24195766 of the University of Zurich, licensed to InCephalo AG. This results in a conflict of interest for coauthors L.S., B.T., M.B., S.D., S.S., T.B., and J.v.B. as co-inventors and/or employees of InCephalo AG. T.B., G.H., M.B., and J.v.B. have equity interests in InCephalo AG. J.v.B. has received speaker fees from Bristol-Myer Squibb. M.A.S. and I.Z. are shareholders of Linkster Therapeutics AG. U.S. has received employee options in Anaveon AG. E.S.W. is an inventor or co-inventor on patents describing antibody repertoire technology (owned by Medical Research Council, UK), half-life extension technology, Abdeg technology, targeting the HER2/HER3 axis, and detection of phosphatidylserine-positive exosomes (owned by UT Southwestern Medical Center or jointly by UT Southwestern Medical Center and argenx BV). E.S.W. is a co-inventor on pending or issued patents describing engineered antibody-drug conjugates and selective depletion of antibodies (jointly owned by UT Southwestern Medical Center and Texas A&M University). E.S.W. has a financial interest in Argenx BV. The remaining authors declare no competing interests.
