## [Transparent Peer Review file · Nature Communications]

FcRn-silencing of IL-12Fc prevents toxicity of local IL-12 therapy and prolongs survival in experimental glioblastoma

Corresponding Author: Dr Johannes vom Berg

Version 0:

Reviewer comments:

Reviewer #1

(Remarks to the Author)

IL-12 immunotherapy for cancer is highly promising, but systemic exposure-induced toxicity remains a major limiting factor for its practical application. Biffinger et al. report a strategy to enhance the retention of intratumorally injected IL-12 in glioblastoma (GBM) by fusing it with an FcRn-silenced Fc fragment. This approach aims to reduce the cytokine's exit from the brain to the bloodstream, thereby achieving therapeutic efficacy within the brain while minimizing systemic exposure to prevent toxicity.

The authors found that FcRn is expressed not only in healthy brain tissue but at even higher levels in GBM. They present convincing data showing that locally delivered FcRn-silenced IL-12Fc has improved retention in brain tissue (including GBM) compared to rIL-12 and wild-type Fc-fused IL-12Fc. This enhanced retention is achieved through two mechanisms: one is increased molecular size, and more importantly, the blockage of FcRn-mediated efflux (transcytosis) from the brain to the blood following local administration.

Furthermore, the authors demonstrate that FcRn-silenced IL-12Fc, both as a monotherapy and in combination with standard treatments (temozolomide chemotherapy and radiotherapy), shows superior local anti-tumor effects in a murine GBM model compared to rIL-12. Notably, the most impressive result is that FcRn-silenced IL-12Fc did not cause overt neurotoxicity. It also protected mice susceptible to systemic IL-12/IFN γ toxicity from severe weight loss and tissue damage, as observed in a C3H/HeJ IFN γ -sensitive mouse model.

Overall, this manuscript presents an innovative strategy to increase the retention of locally-administered IL-12Fc in the brain, thereby reducing systemic toxicity. This approach could be broadly applicable to other central nervous system (CNS) diseases, enabling the restriction of antibody or Fc-fusion protein activity to the brain parenchyma.

However, the following issues should be addressed before the manuscript is accepted for publication in Nature Communications.

Major points

1. Figure 1D-F and Figure 2C: The authors compared the residual levels of IL-12 in the brain and blood under only one condition: a 1 μ g test protein injection with analysis conducted 6 hours after administration. However, the efflux is likely dependent on both the amount of protein and the time after administration. To better understand how dose and time influence the efflux, dose-dependent and time-dependent experiments are necessary.

2. Figure 2E,F and Figure 4: In the extended pharmacokinetic (PK) analysis of the brain and blood in hFcRn Tg32 mice (Figure 2E,F), the differences between IL-12Fc WT and IL-12Fc NHQ do not appear to be substantial. Additionally, in Figure 4, the authors compared the toxicity of rIL-12 and IL-12Fc NHQ, but did not provide data for IL-12Fc WT. Since the central premise of this manuscript is the use of an FcRn-silenced Fc (IL-12Fc NHQ) to reduce or eliminate potential systemic toxicity, a comparison of systemic toxicity between IL-12Fc WT and IL-12Fc NHQ is essential.

3. The authors fused Fc to IL-12 to increase the local half-life in the brain. There are previous studies that have fused Fc to IL-12 to extend serum half-life. Notably, one study (PMID: 29900039) demonstrated that a heterodimeric Fc-fused IL-12 in a monovalent binding format (mono-mIL12-Fc) was more potent than wild-type Fc-based bivalent binding IL-12Fc (bi-mIL12-Fc) for eradicating large, established immunogenic tumors without significant toxicities. This was achieved by enhancing IFN γ production and promoting the proliferation of immune effector cells in tumors. It would be valuable for the authors to compare their bivalent binding IL-12Fc NHQ strategy with the previously reported monovalent binding approach and discuss its impact on T cell phenotypes.

Minor points:

1. Fig. 2D: Please indicate whether the IL-12 levels, following treatment with IL-12Fc WT, with or without MST-HN Fc, are significantly different. If they are not, please explain the reasons.
2. There are many typos. Please check thoroughly the manuscript for typos and grammatical issues.

Reviewer #2

(Remarks to the Author)

In this paper, the authors present a detailed investigation of FcRn-silenced IL-12Fc and its effects on tumor retention, systemic leakage, and combination therapies in glioblastoma models. They explore the molecule's improved brain retention and reduced toxicity while analyzing its synergy with chemotherapy and radiotherapy. The figures provide important visual data supporting these findings, but certain aspects could be clarified for a deeper understanding of the results.

Figure 1:

- Clarify the magnitude and importance of the difference between IL-12Fc and rIL-12 in terms of brain concentration and blood levels. Specifically, how much higher is the brain concentration of IL-12Fc compared to rIL-12, and how much lower are the blood levels? This will help the reader grasp the significance of the findings.
- The results section refers to "early time points," but no information is provided about later time points. Consider adding a sentence to explain what later experiments may reveal or why the early time points are particularly important.

Figure 2:

- the NHQ variant showed significantly better retention than the IAQ, AAA, and WT variants. It would be helpful to quantify how much better NHQ performed compared to the other variants, providing specific fold differences or percentages in text.

Figure 3C:

- Include the number of mice per group and the p-values between groups. Additionally, clarify if mL-12hFc NHQ is significantly different from mL-12hFc WT.

Figure 4:

- While differences in weight loss, tissue damage, and cytokine levels are presented, it would be helpful to include p-values or other statistical measures for key comparisons (e.g., between mL-12 and mL-12hFc-treated animals).

Figure 5:

- While the text mentions that combining IL-12 with TMZ prolongs survival, it would be helpful to explain more clearly how these two treatments work synergistically. For example, how does TMZ-induced antigen release complement IL-12's immune reprogramming? Specifically, testing local immune reprogramming within the tumor microenvironment in the absence of peripheral immune cell influx will clarify whether local or systemic immune responses drive efficacy. This can be achieved by comparing tumor control with and without peripheral immune cell depletion and monitoring local immune cell activation and tumor progression.

Figure 6:

- While the inflammatory activation patterns were conserved across most patients, it would be beneficial to discuss any observed variability in more detail. Were there any outliers or patients who responded differently to IL-12Fc NHQ treatment? Addressing any variability would give a fuller picture of the potential clinical applications.

Reviewer #3

(Remarks to the Author)

Reviewer #4

(Remarks to the Author)

"FcRn-silencing of IL-12Fc prevents toxicity of local IL-12 therapy and prolongs survival in experimental glioblastoma" by Beffinger M et al

What are the noteworthy results?

The authors have engineered a large IL-12Fc fusion cytokine that not only exhibited prolonged retention in the brain but also reduced levels in the circulation reducing toxicity. Murine GBM were used (mostly GL261 but also SB28 in some studies)

along with convection enhanced delivery and showed that FcRn-silenced IL-12Fc induced more durable responses with less systemic exposure and appeared to synergize with radio- and chemotherapy. A FcRn-silenced IL-12Fc NHQ variant showed increased retention in the brain and reduced systemic prevalence upon local intraparenchymal administration, likely due to reduced transcytosis at the BBB.

“Overall, local treatment of established gliomas with all IL-12 variants was associated with increased survival and proinflammatory lymphocyte polarization compared to untreated controls, with FcRn-silenced IL-12Fc additionally exhibiting the highest frequency of durable clinical responses and lowest systemic exposure to IL-12”

In safety studies, they noted substantial alveolar damage, oedema, and lung haemorrhage in animals that received IL-12 and signs of cardiac degeneration in mice given mL-12, but not those treated with mL-12hFc (Fig. 4 C”

They further used a human Blood Brain Barrier model consisting of FcRn-positive iPSC-derived brain microvascular endothelial cells and showed that IL-12Fc NHQ appeared better-retained in the human brain compared to rIL-12 or L-12Fc WT, and was associated with inflammatory polarization.

Significance to the field and related fields:

IL-12 is an important immune stimulant used in oncolytic viruses and other forms of cancer therapy. These findings are important particularly as they relate to intracranial tumors but likely are applicable to other cancers as well.

Does the work support the conclusions and claims, or is additional evidence needed? and data analysis, interpretation and conclusions:

Likely yes overall but would benefit from the following:

FIG 3C. Significance appears to be only tested vs. control, should also show significance of mL-12Fc WT and of mL-12Fc NHQ vs of mL-12

FIG 5B. Significance appears to be only tested vs. control, should also show significance of mL-12Fc NHQ+TMZ vs. IL-12Fc NHQ alone and TMZ alone

FIG 5F. Significance appears to be only tested vs. control, should also show significance of mL-12Fc NHQ+RT vs. IL-12Fc NHQ alone and RT alone

FIG S8B. The same issue applies and significance should be tested between IL-12Fc NHQ +anti-VLA4 vs. IL-12Fc NHQ alone and mL-12 +anti-VLA4

Is the methodology sound?

Yes with additional significance testing as noted above.

Is there enough detail provided in the methods for the work to be reproduced?

Yes

Version 1:

Reviewer comments:

Reviewer #1

(Remarks to the Author)

The additional experiments and detailed revisions have significantly strengthened the manuscript, resolving all critical issues identified earlier. The revised version now meets the standards required for publication, presenting clear, well-supported conclusions and robust scientific evidence.

Reviewer #2

(Remarks to the Author)

The authors answered all my questions

Reviewer #3

(Remarks to the Author)

Reviewer #4

(Remarks to the Author)

No additional comments

NCOMMS-24-51259 submission

“FcRn-silencing of IL-12Fc prevents toxicity of local IL-12 therapy and prolongs survival in experimental glioblastoma”

Beffinger et al, response to external reviews

We thank the Reviewers for their favourable assessment of our work and the constructive comments. We have now performed additional *in vivo* experiments that have further strengthened the manuscript: specifically, we have clearly demonstrated and confirmed the drop in systemic toxicity between IL-12Fc WT vs IL-12Fc NHQ. Throughout the revised manuscript we indicate changes in response to the Reviewers' comments in red ink. Below, please find our point-by-point response to the individual queries raised.

Reviewer #1 (Remarks to the Author): Expert in IL-12 immunotherapy

IL-12 immunotherapy for cancer is highly promising, but systemic exposure-induced toxicity remains a major limiting factor for its practical application. Biffinger et al. report a strategy to enhance the retention of intratumorally injected IL-12 in glioblastoma (GBM) by fusing it with an FcRn-silenced Fc fragment. This approach aims to reduce the cytokine's exit from the brain to the bloodstream, thereby achieving therapeutic efficacy within the brain while minimizing systemic exposure to prevent toxicity.

The authors found that FcRn is expressed not only in healthy brain tissue but at even higher levels in GBM. They present convincing data showing that locally delivered FcRn-silenced IL-12Fc has improved retention in brain tissue (including GBM) compared to rIL-12 and wild-type Fc-fused IL-12Fc. This enhanced retention is achieved through two mechanisms: one is increased molecular size, and more importantly, the blockage of FcRn-mediated efflux (transcytosis) from the brain to the blood following local administration.

Furthermore, the authors demonstrate that FcRn-silenced IL-12Fc, both as a monotherapy and in combination with standard treatments (temozolomide chemotherapy and radiotherapy), shows superior local anti-tumor effects in a murine GBM model compared to rIL-12. Notably, the most impressive result is that FcRn-silenced IL-12Fc did not cause overt neurotoxicity. It also protected mice susceptible to systemic IL-12/IFN γ toxicity from severe weight loss and tissue damage, as observed in a C3H/HeJ IFN γ -sensitive mouse model.

Overall, this manuscript presents an innovative strategy to increase the retention of locally-administered IL-12Fc in the brain, thereby reducing systemic toxicity. This approach could be broadly applicable to other central nervous system (CNS) diseases, enabling the restriction of antibody or Fc-fusion protein activity to the brain parenchyma.

We thank Reviewer #1 for the positive assessment of our work

However, the following issues should be addressed before the manuscript is accepted for publication in Nature Communications.

Major points

1. Figure 1D-F and Figure 2C: The authors compared the residual levels of IL-12 in the brain and blood under only one condition: a 1 μ g test protein injection with analysis conducted 6 hours after administration. However, the efflux is likely dependent on both the amount of protein and the time after administration. To better understand how dose and time influence the efflux, dose-dependent and time-dependent experiments are necessary.

We thank Reviewer #1 for this suggestion, which we have readily incorporated into the revised manuscript:

> Figure 1D-F: We have repeated leakage studies in Tg32 animals with 0.2 \$\mu\$ g as well as with 5 \$\mu\$ g of rIL-12 and IL-12Fc WT and added a 24-hour timepoint to the original six hours. These new data confirm our original conclusions and are included in the revised manuscript as Figure S1 A-D.

In summary, when we repeated the leakage experiment with 0.2 µg (5x less) and with 5 µg (5x more) rIL-12 and IL-12Fc WT, we did not observe any difference in the residual brain amounts of these agents at six hours post CED; though we did see a tendency towards higher tissue retention of IL-12Fc WT after 24 hours. At both timepoints, IL-12 levels in the blood were significantly higher following administration of rIL-12 compared to IL-12Fc WT. However, the fold difference between rIL-12 and IL-12Fc levels in plasma seems to decrease with increasing amounts of injected protein. Overall, and as expected, both local and peripheral levels were higher at the earlier timepoint and if more compound was administered. We thus concluded that our initial observation can be extended to a wider dosing range (from 0.2 µg to 5 µg) as well as to an extended observation window (from six to 24 hours). These new data are described in the main text.

> Figure 2C: Above experiments (Fig. S1 A-D, which now cover a range of 0.2 µg to 5 µg and resulted in comparable results as with 1 µg) demonstrated that our initial observations with 1 µg apply to a wider dosing range. We thus opted to expand the comparison of selected IL-12Fc variants (WT, IAQ, AAA and NHQ) to a 24h timepoint only with 1 µg (which also later serves as a therapeutically relevant dose throughout the manuscript), and observed similar, albeit smaller, statistically non-significant differences between WT and IAQ variants versus AAA and NHQ variants. This additional leakage experiment is now added as supplementary information (Fig. S2 G) and is also mentioned in the main text.

Together with the extended PK analysis of IL-12, IL-12Fc WT and IL-12Fc NHQ beyond 24 hours post administration (Fig. 2 E, F and Fig. S2 I, J) the manuscript now contains comprehensive PK data in FcRn Tg32 mice.

2. Figure 2E,F and Figure 4: In the extended pharmacokinetic (PK) analysis of the brain and blood in hFcRn Tg32 mice (Figure 2E,F), the differences between IL-12Fc WT and IL-12Fc NHQ do not appear to be substantial. Additionally, in Figure 4, the authors compared the toxicity of rIL-12 and IL-12Fc NHQ, but did not provide data for IL-12Fc WT. Since the central premise of this manuscript is the use of an FcRn-silenced Fc (IL-12Fc NHQ) to reduce or eliminate potential systemic toxicity, a comparison of systemic toxicity between IL-12Fc WT and IL-12Fc NHQ is essential.

We thank the Reviewer for the important and valid points raised. Their comments not only led us to re-analyze our data but also to perform additional experiments, which indeed further strengthened and confirmed the premise that FcRn-silenced IL-12Fc exhibits significantly reduced systemic toxicity compared to both rIL-12 and IL-12 Fc WT:

> Figure 2 E,F: We have re-analyzed the extended PK analysis using one-way ANOVA, which confirmed that there was significantly higher concentrations of IL-12 in brain on day 0, and significantly lower concentrations of IL-12 in plasma across days 0-3 following IL-12Fc NHQ compared to rIL-12 treatment. The timepoints where significance is reached (p-value indicated) are now highlighted in the revised version of Fig. 2 E. We also performed statistical comparisons of all three groups for the overall exposure (AUC) using

an unpaired two-tailed t-test, correcting for multiple testing using the Bonferroni method, which confirmed the significance of the plasma data from days 0-3, as detailed in the revised figure and figure legend.

Building on the improved analysis above, we next conducted additional experiments which clearly show that IL-12Fc NHQ rather than IL-12Fc WT is the strongest candidate for treatment of the GBM TME without risk of peripheral toxicity:

The FcRn:Fc interaction both facilitates brain-to-blood export (Schellhammer et al, 2023, *iScience* 26, 108132) and mediates peripheral recycling of IgG, (Roopenian & Akilesh, 2007, *Nat Rev Immunol* 7, 715-725). We thus hypothesized that FcRn-binding of IL-12 Fc WT would counteract its confinement to the brain by prolonging its serum half-life once leaked into the blood.

To test this hypothesis, we adjusted our experimental setup to a continuous intratumoral delivery via osmotic minipumps, as previously reported (vom Berg et al, 2013, *J Exp Med*, 210(13):2803-11). We took advantage of the low affinity of human FcRn (as found in FcRn Tg32 mice) for murine IgG (Ober et al, 2001, *Int Immunol*, 13(12):1551-9) using a murine IL-12Fc version with a murine IgG3 Fc part (as described in Belladonna et al, 2002, *J Immunol*, 168:5448-5454). Compared to WT mice, where the mFcRn:mFc interaction leads to a gradual increase in mIL-12Fc levels, mIL-12Fc levels remained at baseline in FcRn Tg32 mice. This exciting observation provides further support for the importance of abolishing the FcRn:Fc interaction in the context of local, intratumoral administration of Fc-linked therapeutics. Furthermore, it suggests that our findings may also be relevant for continuously administered or *in situ*-produced therapeutics.

This additional leakage experiment is now added as supplementary information (Fig. S1 L) and is described in the main text.

> Figure 4: We also repeated the original study in C3H/HeJ mice with a focus on systemic toxicity of all three molecules: rIL-12, mIL-12hFc WT and mIL-12hFc NHQ. This comprehensive side-by-side analysis confirmed the superiority of FcRn-silenced IL-12 Fc over both IL-12 and IL-12 Fc.

In these experiments, we saw that mIL-12hFc WT behaved similarly to rIL-12 with regards to spleen weight at both 1 µg and 5 µg doses, as well as eliciting comparable levels of IFNγ and CCL-2 in plasma. Importantly, these levels were significantly higher than those following treatment with mIL-12hFc NHQ across a range of timepoints and two different doses. The new results align well with the data in Figures 2 and 3, generated in tumor-naïve FcRn Tg32 mice and tumor-bearing C57BL/6 animals; in addition, these new experiments show that the systemic toxicity elicited in C3H/HeJ animals by mIL-12hFc WT is equally strong as for rIL-12. This experiment is now described in the main text as an initial study focusing on systemic toxicity (Figure S6), prior to a comprehensive pathological examination that focuses on comparing neurotoxicity of mIL-12 and mIL-12hFc NHQ. The entire results of this previous study have now been moved to Figure 4.

3. The authors fused Fc to IL-12 to increase the local half-life in the brain. There are previous studies that have fused Fc to IL-12 to extend serum half-life. Notably, one study

(PMID: 29900039) demonstrated that a heterodimeric Fc-fused IL-12 in a monovalent binding format (mono-mIL12-Fc) was more potent than wild-type Fc-based bivalent binding IL-12Fc (bi-mIL12-Fc) for eradicating large, established immunogenic tumors without significant toxicities. This was achieved by enhancing IFN γ production and promoting the proliferation of immune effector cells in tumors. It would be valuable for the authors to compare their bivalent binding IL-12Fc NHQ strategy with the previously reported monovalent binding approach and discuss its impact on T cell phenotypes.

We thank the Reviewer for this interesting idea. In the mentioned publication (PMID: 29900039, Jung et al Oncoimmunology 2018), Jung and colleagues compared two IL-12Fc variants and indeed the *bi-IL12-Fc* they describe has some resemblance to the IL-12Fc fusion used in our study. While Jung and colleagues do not compare the two IL-12Fc designs to unmodified recombinant IL-12, a later study by Gutierrez and colleagues (Gutierrez et al, 2023, Med 4(5): 326-340 e325) did compare elicited IFN γ levels, which were in the range of recombinant IL-12, in some cases slightly lower. In both cases the dosing was systemic and the elicited prolonged IFN γ levels substantially higher than in our current study.

However, there are a number of important differences in the approach taken by Jung and later by Gutierrez and their coworkers that make a direct and fair comparison difficult: i) engineering efforts were aimed at systemic therapy with a serum half-life extended molecule vs local brain tumor therapy with a serum half-life reduced, locally retained molecule as presented by us; ii) flank tumors and brain tumors differ substantially in their biology and T cell immunology and iii) different administration routes (i.p. vs i.t.) would be expected to have a relevant impact. Most importantly, the high degree of freedom of the IL-12 subunits in Jung et al's *bi-IL12-Fc* due to G₄S linkers between the p35 c-terminus and Fc part may generate a substantially different target engagement than the rigid linkage in our IL-12Fc variants and could trigger a very different T cell phenotype.

While further studies on the mechanism of action of IL-12 Fc fusion molecules are clearly warranted, at this point we believe that the two approaches differ in more ways than IL-12 valency and additional detailed and systematic comparison of *mono-* vs *bi-IL12-Fc* constructs in our local brain tumor treatment setting is beyond the scope of the current manuscript. We thus stressed the local, intratumoral administration and exclusive local mode of action in the Discussion and are excited about an independent in-depth comparison as a future project.

Minor points:

1. Fig. 2D: Please indicate whether the IL-12 levels, following treatment with IL-12Fc WT, with or without MST-HN Fc, are significantly different. If they are not, please explain the reasons.

We thank Reviewer for bringing ways to enhance the description of this experiment to our attention. Upon further statistical analysis, we can confirm that the observed difference between the IL-12Fc WT samples with and without an FcRn blocker (MST-HN Fc), as depicted in Figure 2D, is indeed statistically significant. This conclusion is supported by a two-tailed unpaired t-test, which yielded a p-value of 0.0278, indicating a statistically

significant level of confidence in the observed results. However, the primary focus of this experiment was to determine whether we could phenocopy the effects of NHQ substitution with an exogenous FcRn blocker. In the absence of MST-HN Fc, IL-12Fc WT brain levels are significantly lower than IL-12Fc NHQ levels ($p=0.0177$). Conversely, in combination with MST-HN Fc, the difference in brain levels is no longer significant ($p=0.8663$) and the levels of IL-12Fc WT increased to those seen for IL-12Fc NHQ. In combination, this indicates sufficient blockade of the Fc:FcRn interaction by MST-HN Fc and thus enhances our understanding of the mechanism leading to the observed PK properties of IL-12Fc NHQ. Taking these into account, we have opted to update our statistical analysis on the two comparator groups.

2. There are many typos. Please check thoroughly the manuscript for typos and grammatical issues.

We thank Reviewer for carefully reading our manuscript and have diligently checked and corrected typos and grammar.

Reviewer #2 (Remarks to the Author): Expert in IL-12 immunotherapy, glioblastoma orthotopic mouse models and therapy

In this paper, the authors present a detailed investigation of FcRn-silenced IL-12Fc and its effects on tumor retention, systemic leakage, and combination therapies in glioblastoma models. They explore the molecule's improved brain retention and reduced toxicity while analyzing its synergy with chemotherapy and radiotherapy. The figures provide important visual data supporting these findings, but certain aspects could be clarified for a deeper understanding of the results.

We thank Reviewer #2 for the positive assessment of our work and the many suggestions for improvement of our manuscript.

Figure 1:

- Clarify the magnitude and importance of the difference between IL-12Fc and rIL-12 in terms of brain concentration and blood levels. Specifically, how much higher is the brain concentration of IL-12Fc compared to rIL-12, and how much lower are the blood levels? This will help the reader grasp the significance of the findings.

We thank the Reviewer for this excellent suggestion and have now incorporated fold-change throughout the description of leakage experiments described in Figure 1, and for the additional experiments now described in Figure S1 A-D, but also in later fragments of the manuscript.

- The results section refers to "early time points," but no information is provided about later time points. Consider adding a sentence to explain what later experiments may reveal or why the early time points are particularly important.

We thank the Reviewer for pointing this out; this concern was also raised by Reviewer #1 and we have accordingly extended our leakage analysis to include different doses as well as a later (24 hour) timepoint. The new data strengthens our initial conclusions and is now included in Figure S1 A-D as repeated leakage studies in Tg32 animals with 0.2 µg as well as with 5 µg of rIL-12 and IL-12Fc WT at 6 hours and 24 hours. Overall, and as expected, both local and peripheral levels were higher at the earlier timepoint and if more compound was administered. We concluded that our initial observation can be extended to a wider dosing range as well as to an extended observation window. These conclusions are described in the main text.

Figure 2:

- the NHQ variant showed significantly better retention than the IAQ, AAA, and WT variants. It would be helpful to quantify how much better NHQ performed compared to

the other variants, providing specific fold differences or percentages in text.

We thank the Reviewer for this suggestion. Also in line with the comments regarding Figure 1, we now provide additional information on brain retention and leakage of variants throughout the description of the most important results from Figure 2.

Figure 3C:

- Include the number of mice per group and the p-values between groups. Additionally, clarify if mIL-12hFc NHQ is significantly different from mIL-12hFc WT.

We thank the Reviewer for this suggestion and now have indicated the p-values for all group comparisons in Figure 3C and further figures; we have also provided the numbers of mice per group in the figure legend.

Figure 4:

- While differences in weight loss, tissue damage, and cytokine levels are presented, it would be helpful to include p-values or other statistical measures for key comparisons (e.g., between mIL-12 and mIL-12hFc-treated animals).

This is an excellent suggestion which we have readily incorporated into the revised Figure 4 and into Figure S6 which describes an additional toxicity experiment: Key readouts such as spleen weight, cytokine levels (IL-12, IFN γ and CCL-2 - now Figure 4, B, D, F & H; and Figure S6 C, E, G & I) and alanine transaminase levels (ALT, Figure S6 J) have now been quantified and statistically assessed. P-values are indicated where $p < 0.05$.

Figure 5:

- While the text mentions that combining IL-12 with TMZ prolongs survival, it would be helpful to explain more clearly how these two treatments work synergistically. For example, how does TMZ-induced antigen release complement IL-12's immune reprogramming? Specifically, testing local immune reprogramming within the tumor microenvironment in the absence of peripheral immune cell influx will clarify whether local or systemic immune responses drive efficacy. This can be achieved by comparing tumor control with and without peripheral immune cell depletion and monitoring local immune cell activation and tumor progression.

We thank the Reviewer for raising this interesting and important topic. At this point we assume that the additive effect derives from chemotherapy's direct cytotoxicity triggering increased antigen release, and reactivation of existing tumor-infiltrating T cells stimulated by IL-12 therapy. While Mathios and colleagues (Mathios et al, 2016, Sci. Transl. Med. 8, 370ra180) do describe such an effect in the context of local chemotherapy and systemic immunotherapy (and not both systemic treatments), it appears that in the local IL-12 therapy setting, also systemic chemotherapy sustains efficacy.

To further define the effects we observed in our efficacy study, we analyzed TILs in TMZ+/- mIL-12hFc NHQ treated animals. Based on the Reviewer's suggestion, we leveraged the fact that TMZ treatment leads to severe lymphopenia and thus a situation where the contribution of the peripheral immune system is compromised. In this setting we observed increased numbers of proliferating *bona fide* resident CD4⁺ cells, the presence of CD8⁺ resident cells and activation/increased antigen presentation by microglia. These observations help to put our observations in a broader perspective of reactivation of TIL vs recruitment of naïve lymphocytes to the treatment site and thus substantially improve the manuscript. The data are now presented in Figure S7 and Figure 5 D-G and further picked up in the discussion section of the revised manuscript.

Figure 6:

- While the inflammatory activation patterns were conserved across most patients, it would be beneficial to discuss any observed variability in more detail. Were there any outliers or patients who responded differently to IL-12Fc NHQ treatment? Addressing any variability would give a fuller picture of the potential clinical applications.

We thank the Reviewer for this important point. Patient to patient variability is expected, however, a comprehensive assessment of the overall heterogeneity of the responses to IL-12Fc NHQ will require a substantially larger cohort to shape future clinical expectations. In our experiments, samples of the earlier pilot series showed lower overall signal intensity and may have exaggerated the impression of a heterogenous response. This relationship is now illustrated in Figure S11 A with a plot of the interquartile range vs sample mean of the collapsed bioreactor samples. In combination with the principal component analysis (Figure S11 B) and the estimated quality weights (Figure S11 C) the context of response variability and sample quality has hopefully become clearer. In addition to these adjustments to Figure S11, we also address these points in the discussion. In addition, we have further clarified the patient sample metadata in the methods section and Table S3.

Reviewer #3 (Remarks to the Author): Early-Career Researcher co-reviewer

We thank Reviewer #3 for critically reviewing our work.

Reviewer #4 (Remarks to the Author): Expert in IL-12 immunotherapy, glioblastoma orthotopic mouse models and therapy

"FcRn-silencing of IL-12Fc prevents toxicity of local IL-12 therapy and prolongs survival in experimental glioblastoma" by Beffinger M et al

What are the noteworthy results?

The authors have engineered a large IL-12Fc fusion cytokine that not only exhibited prolonged retention in the brain but also reduced levels in the circulation reducing toxicity. Murine GBM were used (mostly GL261 but also SB28 in some studies) along with convection enhanced delivery and showed that FcRn-silenced IL-12Fc induced more durable responses with less systemic exposure and appeared to synergize with radio- and chemotherapy. A FcRn-silenced IL-12Fc NHQ variant showed increased retention in the brain and reduced systemic prevalence upon local intraparenchymal administration, likely due to reduced transcytosis at the BBB.

"Overall, local treatment of established gliomas with all IL-12 variants was associated with increased survival and proinflammatory lymphocyte polarization compared to untreated controls, with FcRn-silenced IL-12Fc additionally exhibiting the highest frequency of durable clinical responses and lowest systemic exposure to IL-12"

In safety studies, they noted substantial alveolar damage, oedema, and lung haemorrhage in animals that received IL-12 and signs of cardiac degeneration in mice given mL-12, but not those treated with mL-12hFc (Fig. 4 C"

They further used a human Blood Brain Barrier model consisting of FcRn-positive iPSC-derived brain microvascular endothelial cells and showed that IL-12Fc NHQ appeared better-retained in the human brain compared to rIL-12 or L-12Fc WT, and was associated with inflammatory polarization.

We thank Reviewer #4 for the positive assessment of our work and the suggestions for improvement of our manuscript.

Significance to the field and related fields:

IL-12 is an important immune stimulant used in oncolytic viruses and other forms of cancer therapy. These findings are important particularly as they relate to intracranial tumors but likely are applicable to other cancers as well.

Does the work support the conclusions and claims, or is additional evidence needed? and data analysis, interpretation and conclusions:

Likely yes overall but would benefit from the following:

FIG 3C. Significance appears to be only tested vs. control, should also show significance of mL-12Fc WT and of mL-12Fc NHQ vs of mL-12

We thank the Reviewer for this comment. We have now modified the figures to include all results of the statistical testing for all groups of the survival analysis: significance values

in Figure 3C include mL-12Fc WT vs mL-12 p=0.1130 and mL-12Fc NHQ vs mL-12 p=0.1489.

FIG 5B. Significance appears to be only tested vs. control, should also show significance of mL-12Fc NHQ+TMZ vs. IL-12Fc NHQ alone and TMZ alone

In Figure 5 B we now show results of significance analysis for all groups, in particular mL-12Fc NHQ + TMZ vs mL-12Fc NHQ p=0.0059, mL-12Fc NHQ + TMZ vs TMZ p=0.0584 and mL-12Fc NHQ vs TMZ p=0.4015

FIG 5F. Significance appears to be only tested vs. control, should also show significance of mL-12Fc NHQ+RT vs. IL-12Fc NHQ alone and RT alone

For Figure 5 F (Figure 5 J in the updated version of the manuscript) we now show results of additional significances for all groups, in particular IL-12Fc NHQ + RT vs IL-12Fc NHQ p=0.0418, IL-12Fc NHQ + RT vs RT p=0.3848 and IL-12Fc NHQ vs RT p=0.6295.

FIG S8B. The same issue applies and significance should be tested between IL-12Fc NHQ +anti-VLA4 vs. IL-12Fc NHQ alone and mL-12 +anti-VLA4

We thank the Reviewer for this comment; also for Figure S8 B we now show statistical analysis for all group comparisons, IL-12Fc NHQ + aVLA4 vs IL-12Fc NHQ p=0.6139 and IL-12Fc NHQ + aVLA4 vs IL-12 + aVLA4 p=0.7326

Is the methodology sound?

Yes with additional significance testing as noted above.

Is there enough detail provided in the methods for the work to be reproduced?

Yes

Reviewer #1 (Remarks to the Author): Expert in IL-12 immunotherapy

IL-12 immunotherapy for cancer is highly promising, but systemic exposure-induced toxicity remains a major limiting factor for its practical application. Biffinger et al. report a strategy to enhance the retention of intratumorally injected IL-12 in glioblastoma (GBM) by fusing it with an FcRn-silenced Fc fragment. This approach aims to reduce the cytokine's exit from the brain to the bloodstream, thereby achieving therapeutic efficacy within the brain while minimizing systemic exposure to prevent toxicity.

The authors found that FcRn is expressed not only in healthy brain tissue but at even higher levels in GBM. They present convincing data showing that locally delivered FcRn-silenced IL-12Fc has improved retention in brain tissue (including GBM) compared to rIL-12 and wild-type Fc-fused IL-12Fc. This enhanced retention is achieved through two mechanisms: one is increased molecular size, and more importantly, the blockage of FcRn-mediated efflux (transcytosis) from the brain to the blood following local administration.

Furthermore, the authors demonstrate that FcRn-silenced IL-12Fc, both as a monotherapy and in combination with standard treatments (temozolomide chemotherapy and radiotherapy), shows superior local anti-tumor effects in a murine GBM model compared to rIL-12. Notably, the most impressive result is that FcRn-silenced IL-12Fc did not cause overt neurotoxicity. It also protected mice susceptible to systemic IL-12/IFN γ toxicity from severe weight loss and tissue damage, as observed in a C3H/HeJ IFN γ -sensitive mouse model.

Overall, this manuscript presents an innovative strategy to increase the retention of locally-administered IL-12Fc in the brain, thereby reducing systemic toxicity. This approach could be broadly applicable to other central nervous system (CNS) diseases, enabling the restriction of antibody or Fc-fusion protein activity to the brain parenchyma.

We thank Reviewer #1 for the positive assessment of our work

However, the following issues should be addressed before the manuscript is accepted for publication in Nature Communications.

Major points

1. Figure 1D-F and Figure 2C: The authors compared the residual levels of IL-12 in the brain and blood under only one condition: a 1 μ g test protein injection with analysis conducted 6 hours after administration. However, the efflux is likely dependent on both the amount of protein and the time after administration. To better understand how dose and time influence the efflux, dose-dependent and time-dependent experiments are necessary.

We thank Reviewer #1 for this suggestion, which we have readily incorporated into the revised manuscript:

> Figure 1D-F: We have repeated leakage studies in Tg32 animals with 0.2 \$\mu\$ g as well as with 5 \$\mu\$ g of rIL-12 and IL-12Fc WT and added a 24-hour timepoint to the original six hours. These new data confirm our original conclusions and are included in the revised manuscript as Figure S1 A-D.

In summary, when we repeated the leakage experiment with 0.2 µg (5x less) and with 5 µg (5x more) rIL-12 and IL-12Fc WT, we did not observe any difference in the residual brain amounts of these agents at six hours post CED; though we did see a tendency towards higher tissue retention of IL-12Fc WT after 24 hours. At both timepoints, IL-12 levels in the blood were significantly higher following administration of rIL-12 compared to IL-12Fc WT. However, the fold difference between rIL-12 and IL-12Fc levels in plasma seems to decrease with increasing amounts of injected protein. Overall, and as expected, both local and peripheral levels were higher at the earlier timepoint and if more compound was administered. We thus concluded that our initial observation can be extended to a wider dosing range (from 0.2 µg to 5 µg) as well as to an extended observation window (from six to 24 hours). These new data are described in the main text.

> Figure 2C: Above experiments (Fig. S1 A-D, which now cover a range of 0.2 µg to 5 µg and resulted in comparable results as with 1 µg) demonstrated that our initial observations with 1 µg apply to a wider dosing range. We thus opted to expand the comparison of selected IL-12Fc variants (WT, IAQ, AAA and NHQ) to a 24h timepoint only with 1 µg (which also later serves as a therapeutically relevant dose throughout the manuscript), and observed similar, albeit smaller, statistically non-significant differences between WT and IAQ variants versus AAA and NHQ variants. This additional leakage experiment is now added as supplementary information (Fig. S2 G) and is also mentioned in the main text.

Together with the extended PK analysis of IL-12, IL-12Fc WT and IL-12Fc NHQ beyond 24 hours post administration (Fig. 2 E, F and Fig. S2 I, J) the manuscript now contains comprehensive PK data in FcRn Tg32 mice.

2. Figure 2E,F and Figure 4: In the extended pharmacokinetic (PK) analysis of the brain and blood in hFcRn Tg32 mice (Figure 2E,F), the differences between IL-12Fc WT and IL-12Fc NHQ do not appear to be substantial. Additionally, in Figure 4, the authors compared the toxicity of rIL-12 and IL-12Fc NHQ, but did not provide data for IL-12Fc WT. Since the central premise of this manuscript is the use of an FcRn-silenced Fc (IL-12Fc NHQ) to reduce or eliminate potential systemic toxicity, a comparison of systemic toxicity between IL-12Fc WT and IL-12Fc NHQ is essential.

We thank the Reviewer for the important and valid points raised. Their comments not only led us to re-analyze our data but also to perform additional experiments, which indeed further strengthened and confirmed the premise that FcRn-silenced IL-12Fc exhibits significantly reduced systemic toxicity compared to both rIL-12 and IL-12 Fc WT:

> Figure 2 E,F: We have re-analyzed the extended PK analysis using one-way ANOVA, which confirmed that there was significantly higher concentrations of IL-12 in brain on day 0, and significantly lower concentrations of IL-12 in plasma across days 0-3 following IL-12Fc NHQ compared to rIL-12 treatment. The timepoints where significance is reached (p-value indicated) are now highlighted in the revised version of Fig. 2 E. We also performed statistical comparisons of all three groups for the overall exposure (AUC) using

an unpaired two-tailed t-test, correcting for multiple testing using the Bonferroni method, which confirmed the significance of the plasma data from days 0-3, as detailed in the revised figure and figure legend.

Building on the improved analysis above, we next conducted additional experiments which clearly show that IL-12Fc NHQ rather than IL-12Fc WT is the strongest candidate for treatment of the GBM TME without risk of peripheral toxicity:

The FcRn:Fc interaction both facilitates brain-to-blood export (Schellhammer et al, 2023, *iScience* 26, 108132) and mediates peripheral recycling of IgG, (Roopenian & Akilesh, 2007, *Nat Rev Immunol* 7, 715-725). We thus hypothesized that FcRn-binding of IL-12 Fc WT would counteract its confinement to the brain by prolonging its serum half-life once leaked into the blood.

To test this hypothesis, we adjusted our experimental setup to a continuous intratumoral delivery via osmotic minipumps, as previously reported (vom Berg et al, 2013, *J Exp Med*, 210(13):2803-11). We took advantage of the low affinity of human FcRn (as found in FcRn Tg32 mice) for murine IgG (Ober et al, 2001, *Int Immunol*, 13(12):1551-9) using a murine IL-12Fc version with a murine IgG3 Fc part (as described in Belladonna et al, 2002, *J Immunol*, 168:5448-5454). Compared to WT mice, where the mFcRn:mFc interaction leads to a gradual increase in mIL-12Fc levels, mIL-12Fc levels remained at baseline in FcRn Tg32 mice. This exciting observation provides further support for the importance of abolishing the FcRn:Fc interaction in the context of local, intratumoral administration of Fc-linked therapeutics. Furthermore, it suggests that our findings may also be relevant for continuously administered or *in situ*-produced therapeutics.

This additional leakage experiment is now added as supplementary information (Fig. S1 L) and is described in the main text.

> Figure 4: We also repeated the original study in C3H/HeJ mice with a focus on systemic toxicity of all three molecules: rmlL-12, mL-12hFc WT and mL-12hFc NHQ. This comprehensive side-by-side analysis confirmed the superiority of FcRn-silenced IL-12 Fc over both IL-12 and IL-12 Fc.

In these experiments, we saw that mL-12hFc WT behaved similarly to rmlL-12 with regards to spleen weight at both 1 µg and 5 µg doses, as well as eliciting comparable levels of IFN γ and CCL-2 in plasma. Importantly, these levels were significantly higher than those following treatment with mL-12hFc NHQ across a range of timepoints and two different doses. The new results align well with the data in Figures 2 and 3, generated in tumor-naïve FcRn Tg32 mice and tumor-bearing C57BL/6 animals; in addition, these new experiments show that the systemic toxicity elicited in C3H/HeJ animals by mL-12hFc WT is equally strong as for rmlL-12. This experiment is now described in the main text as an initial study focusing on systemic toxicity (Figure S6), prior to a comprehensive pathological examination that focuses on comparing neurotoxicity of mL-12 and mL-12hFc NHQ. The entire results of this previous study have now been moved to Figure 4.

3. The authors fused Fc to IL-12 to increase the local half-life in the brain. There are previous studies that have fused Fc to IL-12 to extend serum half-life. Notably, one study

(PMID: 29900039) demonstrated that a heterodimeric Fc-fused IL-12 in a monovalent binding format (*mono-mIL12-Fc*) was more potent than wild-type Fc-based bivalent binding IL-12Fc (*bi-mIL12-Fc*) for eradicating large, established immunogenic tumors without significant toxicities. This was achieved by enhancing IFN γ production and promoting the proliferation of immune effector cells in tumors. It would be valuable for the authors to compare their bivalent binding IL-12Fc NHQ strategy with the previously reported monovalent binding approach and discuss its impact on T cell phenotypes.

We thank the Reviewer for this interesting idea. In the mentioned publication (PMID: 29900039, Jung et al *Oncoimmunology* 2018), Jung and colleagues compared two IL-12Fc variants and indeed the *bi-IL12-Fc* they describe has some resemblance to the IL-12Fc fusion used in our study. While Jung and colleagues do not compare the two IL-12Fc designs to unmodified recombinant IL-12, a later study by Gutierrez and colleagues (Gutierrez et al, 2023, *Med* 4(5): 326-340 e325) did compare elicited IFN γ levels, which were in the range of recombinant IL-12, in some cases slightly lower. In both cases the dosing was systemic and the elicited prolonged IFN γ levels substantially higher than in our current study.

However, there are a number of important differences in the approach taken by Jung and later by Gutierrez and their coworkers that make a direct and fair comparison difficult: i) engineering efforts were aimed at systemic therapy with a serum half-life extended molecule vs local brain tumor therapy with a serum half-life reduced, locally retained molecule as presented by us; ii) flank tumors and brain tumors differ substantially in their biology and T cell immunology and iii) different administration routes (i.p. vs i.t.) would be expected to have a relevant impact. Most importantly, the high degree of freedom of the IL-12 subunits in Jung et al's *bi-IL12-Fc* due to G₄S linkers between the p35 c-terminus and Fc part may generate a substantially different target engagement than the rigid linkage in our IL-12Fc variants and could trigger a very different T cell phenotype.

While further studies on the mechanism of action of IL-12 Fc fusion molecules are clearly warranted, at this point we believe that the two approaches differ in more ways than IL-12 valency and additional detailed and systematic comparison of *mono-* vs *bi-IL12-Fc* constructs in our local brain tumor treatment setting is beyond the scope of the current manuscript. We thus stressed the local, intratumoral administration and exclusive local mode of action in the Discussion and are excited about an independent in-depth comparison as a future project.

Minor points:

1. Fig. 2D: Please indicate whether the IL-12 levels, following treatment with IL-12Fc WT, with or without MST-HN Fc, are significantly different. If they are not, please explain the reasons.

We thank Reviewer for bringing ways to enhance the description of this experiment to our attention. Upon further statistical analysis, we can confirm that the observed difference between the IL-12Fc WT samples with and without an FcRn blocker (MST-HN Fc), as depicted in Figure 2D, is indeed statistically significant. This conclusion is supported by a two-tailed unpaired t-test, which yielded a p-value of 0.0278, indicating a statistically

significant level of confidence in the observed results. However, the primary focus of this experiment was to determine whether we could phenocopy the effects of NHQ substitution with an exogenous FcRn blocker. In the absence of MST-HN Fc, IL-12Fc WT brain levels are significantly lower than IL-12Fc NHQ levels ($p=0.0177$). Conversely, in combination with MST-HN Fc, the difference in brain levels is no longer significant ($p=0.8663$) and the levels of IL-12Fc WT increased to those seen for IL-12Fc NHQ. In combination, this indicates sufficient blockade of the Fc:FcRn interaction by MST-HN Fc and thus enhances our understanding of the mechanism leading to the observed PK properties of IL-12Fc NHQ. Taking these into account, we have opted to update our statistical analysis on the two comparator groups.

2. There are many typos. Please check thoroughly the manuscript for typos and grammatical issues.

We thank Reviewer for carefully reading our manuscript and have diligently checked and corrected typos and grammar.

Reviewer #2 (Remarks to the Author): Expert in IL-12 immunotherapy, glioblastoma orthotopic mouse models and therapy

In this paper, the authors present a detailed investigation of FcRn-silenced IL-12Fc and its effects on tumor retention, systemic leakage, and combination therapies in glioblastoma models. They explore the molecule's improved brain retention and reduced toxicity while analyzing its synergy with chemotherapy and radiotherapy. The figures provide important visual data supporting these findings, but certain aspects could be clarified for a deeper understanding of the results.

We thank Reviewer #2 for the positive assessment of our work and the many suggestions for improvement of our manuscript.

Figure 1:

- Clarify the magnitude and importance of the difference between IL-12Fc and rIL-12 in terms of brain concentration and blood levels. Specifically, how much higher is the brain concentration of IL-12Fc compared to rIL-12, and how much lower are the blood levels? This will help the reader grasp the significance of the findings.

We thank the Reviewer for this excellent suggestion and have now incorporated fold-change throughout the description of leakage experiments described in Figure 1, and for the additional experiments now described in Figure S1 A-D, but also in later fragments of the manuscript.

- The results section refers to "early time points," but no information is provided about later time points. Consider adding a sentence to explain what later experiments may reveal or why the early time points are particularly important.

We thank the Reviewer for pointing this out; this concern was also raised by Reviewer #1 and we have accordingly extended our leakage analysis to include different doses as well as a later (24 hour) timepoint. The new data strengthens our initial conclusions and is now included in Figure S1 A-D as repeated leakage studies in Tg32 animals with 0.2 µg as well as with 5 µg of rIL-12 and IL-12Fc WT at 6 hours and 24 hours. Overall, and as expected, both local and peripheral levels were higher at the earlier timepoint and if more compound was administered. We concluded that our initial observation can be extended to a wider dosing range as well as to an extended observation window. These conclusions are described in the main text.

Figure 2:

- the NHQ variant showed significantly better retention than the IAQ, AAA, and WT variants. It would be helpful to quantify how much better NHQ performed compared to

the other variants, providing specific fold differences or percentages in text.

We thank the Reviewer for this suggestion. Also in line with the comments regarding Figure 1, we now provide additional information on brain retention and leakage of variants throughout the description of the most important results from Figure 2.

Figure 3C:

- Include the number of mice per group and the p-values between groups. Additionally, clarify if mIL-12hFc NHQ is significantly different from mIL-12hFc WT.

We thank the Reviewer for this suggestion and now have indicated the p-values for all group comparisons in Figure 3C and further figures; we have also provided the numbers of mice per group in the figure legend.

Figure 4:

- While differences in weight loss, tissue damage, and cytokine levels are presented, it would be helpful to include p-values or other statistical measures for key comparisons (e.g., between mIL-12 and mIL-12hFc-treated animals).

This is an excellent suggestion which we have readily incorporated into the revised Figure 4 and into Figure S6 which describes an additional toxicity experiment: Key readouts such as spleen weight, cytokine levels (IL-12, IFN γ and CCL-2 - now Figure 4, B, D, F & H; and Figure S6 C, E, G & I) and alanine transaminase levels (ALT, Figure S6 J) have now been quantified and statistically assessed. P-values are indicated where $p < 0.05$.

Figure 5:

- While the text mentions that combining IL-12 with TMZ prolongs survival, it would be helpful to explain more clearly how these two treatments work synergistically. For example, how does TMZ-induced antigen release complement IL-12's immune reprogramming? Specifically, testing local immune reprogramming within the tumor microenvironment in the absence of peripheral immune cell influx will clarify whether local or systemic immune responses drive efficacy. This can be achieved by comparing tumor control with and without peripheral immune cell depletion and monitoring local immune cell activation and tumor progression.

We thank the Reviewer for raising this interesting and important topic. At this point we assume that the additive effect derives from chemotherapy's direct cytotoxicity triggering increased antigen release, and reactivation of existing tumor-infiltrating T cells stimulated by IL-12 therapy. While Mathios and colleagues (Mathios et al, 2016, Sci. Transl. Med. 8, 370ra180) do describe such an effect in the context of local chemotherapy and systemic immunotherapy (and not both systemic treatments), it appears that in the local IL-12 therapy setting, also systemic chemotherapy sustains efficacy.

To further define the effects we observed in our efficacy study, we analyzed TILs in TMZ+/- mIL-12hFc NHQ treated animals. Based on the Reviewer's suggestion, we leveraged the fact that TMZ treatment leads to severe lymphopenia and thus a situation where the contribution of the peripheral immune system is compromised. In this setting we observed increased numbers of proliferating *bona fide* resident CD4⁺ cells, the presence of CD8⁺ resident cells and activation/increased antigen presentation by microglia. These observations help to put our observations in a broader perspective of reactivation of TIL vs recruitment of naïve lymphocytes to the treatment site and thus substantially improve the manuscript. The data are now presented in Figure S7 and Figure 5 D-G and further picked up in the discussion section of the revised manuscript.

Figure 6:

- While the inflammatory activation patterns were conserved across most patients, it would be beneficial to discuss any observed variability in more detail. Were there any outliers or patients who responded differently to IL-12Fc NHQ treatment? Addressing any variability would give a fuller picture of the potential clinical applications.

We thank the Reviewer for this important point. Patient to patient variability is expected, however, a comprehensive assessment of the overall heterogeneity of the responses to IL-12Fc NHQ will require a substantially larger cohort to shape future clinical expectations. In our experiments, samples of the earlier pilot series showed lower overall signal intensity and may have exaggerated the impression of a heterogenous response. This relationship is now illustrated in Figure S11 A with a plot of the interquartile range vs sample mean of the collapsed bioreactor samples. In combination with the principal component analysis (Figure S11 B) and the estimated quality weights (Figure S11 C) the context of response variability and sample quality has hopefully become clearer. In addition to these adjustments to Figure S11, we also address these points in the discussion. In addition, we have further clarified the patient sample metadata in the methods section and Table S3.

Reviewer #3 (Remarks to the Author): Early-Career Researcher co-reviewer

We thank Reviewer #3 for critically reviewing our work.

Reviewer #4 (Remarks to the Author): Expert in IL-12 immunotherapy, glioblastoma orthotopic mouse models and therapy

"FcRn-silencing of IL-12Fc prevents toxicity of local IL-12 therapy and prolongs survival in experimental glioblastoma" by Beffinger M et al

What are the noteworthy results?

The authors have engineered a large IL-12Fc fusion cytokine that not only exhibited prolonged retention in the brain but also reduced levels in the circulation reducing toxicity. Murine GBM were used (mostly GL261 but also SB28 in some studies) along with convection enhanced delivery and showed that FcRn-silenced IL-12Fc induced more durable responses with less systemic exposure and appeared to synergize with radio- and chemotherapy. A FcRn-silenced IL-12Fc NHQ variant showed increased retention in the brain and reduced systemic prevalence upon local intraparenchymal administration, likely due to reduced transcytosis at the BBB.

"Overall, local treatment of established gliomas with all IL-12 variants was associated with increased survival and proinflammatory lymphocyte polarization compared to untreated controls, with FcRn-silenced IL-12Fc additionally exhibiting the highest frequency of durable clinical responses and lowest systemic exposure to IL-12"

In safety studies, they noted substantial alveolar damage, oedema, and lung haemorrhage in animals that received IL-12 and signs of cardiac degeneration in mice given mL-12, but not those treated with mL-12hFc (Fig. 4 C"

They further used a human Blood Brain Barrier model consisting of FcRn-positive iPSC-derived brain microvascular endothelial cells and showed that IL-12Fc NHQ appeared better-retained in the human brain compared to rIL-12 or L-12Fc WT, and was associated with inflammatory polarization.

We thank Reviewer #4 for the positive assessment of our work and the suggestions for improvement of our manuscript.

Significance to the field and related fields:

IL-12 is an important immune stimulant used in oncolytic viruses and other forms of cancer therapy. These findings are important particularly as they relate to intracranial tumors but likely are applicable to other cancers as well.

Does the work support the conclusions and claims, or is additional evidence needed? and data analysis, interpretation and conclusions:

Likely yes overall but would benefit from the following:

FIG 3C. Significance appears to be only tested vs. control, should also show significance of mL-12Fc WT and of mL-12Fc NHQ vs of mL-12

We thank the Reviewer for this comment. We have now modified the figures to include all results of the statistical testing for all groups of the survival analysis: significance values

in Figure 3C include mL-12Fc WT vs mL-12 p=0.1130 and mL-12Fc NHQ vs mL-12 p=0.1489.

FIG 5B. Significance appears to be only tested vs. control, should also show significance of mL-12Fc NHQ+TMZ vs. IL-12Fc NHQ alone and TMZ alone

In Figure 5 B we now show results of significance analysis for all groups, in particular mL-12Fc NHQ + TMZ vs mL-12Fc NHQ p=0.0059, mL-12Fc NHQ + TMZ vs TMZ p=0.0584 and mL-12Fc NHQ vs TMZ p=0.4015

FIG 5F. Significance appears to be only tested vs. control, should also show significance of mL-12Fc NHQ+RT vs. IL-12Fc NHQ alone and RT alone

For Figure 5 F (Figure 5 J in the updated version of the manuscript) we now show results of additional significances for all groups, in particular IL-12Fc NHQ + RT vs IL-12Fc NHQ p=0.0418, IL-12Fc NHQ + RT vs RT p=0.3848 and IL-12Fc NHQ vs RT p=0.6295.

FIG S8B. The same issue applies and significance should be tested between IL-12Fc NHQ +anti-VLA4 vs. IL-12Fc NHQ alone and mL-12 +anti-VLA4

We thank the Reviewer for this comment; also for Figure S8 B we now show statistical analysis for all group comparisons, IL-12Fc NHQ + aVLA4 vs IL-12Fc NHQ p=0.6139 and IL-12Fc NHQ + aVLA4 vs IL-12 + aVLA4 p=0.7326

Is the methodology sound?

Yes with additional significance testing as noted above.

Is there enough detail provided in the methods for the work to be reproduced?

Yes